# Variability in Above and Belowground Carbon Stocks in a Siberian Larch Watershed

Elizabeth E. Webb
        Woods Hole Research Center, 149 Woods Hole Road, Falmouth, MA, 02540

Kathryn Heard
        Western Washington University, 516 High Street, Bellingham, WA 98225
         Bellingham, Washington

Susan M. Natali*
        Woods Hole Research Center, 149 Woods Hole Road, Falmouth, MA, 02540

Andrew G. Bunn
        Department of Environmental Science, Western Washington University, 516 High
        Street, Bellingham, WA 98225

Heather D. Alexander
        Department of Forestry, Forest and Wildlife Research Center, Mississippi State
        University, MS 39762

Logan T. Berner
        School of Informatics, Computing, and Cyber Systems, Northern Arizona
        University, Flagstaff, AZ 86011

Alexander Kholodov
        University of Alaska, 903 Koyukuk Dr., Fairbanks, AK, 99775
        Institute of Physical-Chemical and Biological Problems of Soil Science RAS, 2
        Institutskaya str., Pushchino, Russia.

Michael M. Loranty
        Department of Geography, Colgate University, 13 Oak Dr, Hamilton, New York
        13346

John D. Schade
        Woods Hole Research Center, 149 Woods Hole Road, Falmouth, MA, 02540

Valentin Spektor,
        Melnikov Permafrost Institute, Siberian Branch of the Russian Academy of
        Sciences, Yakutsk, Siberia

Nikita Zimov
        Northeast Science Station, Cherskiy, Russia

*Corresponding author.  Please direct correspondence to snatali@whrc.org

**ABSTRACT**

2       Permafrost soils store between 1,330-1,580 Pg carbon (C), which is three times the

amount of C in global vegetation, almost twice the amount of C in the atmosphere, and half of
the global soil organic C pool.  Despite the massive amount of C in permafrost, estimates of soil
C storage in the high latitude permafrost region are highly uncertain, primarily due to under
sampling at all spatial scales; circumpolar soil C estimates lack sufficient continental spatial
diversity, regional intensity, and replication at the field-site level.  Siberian forests are
particularly under sampled, yet the larch forests that dominate this region may store more than
twice as much soil C as all other boreal forest types in the continuous permafrost zone combined.
Here we present above and belowground C stocks from twenty sites representing a gradient of
stand age and structure in a larch watershed of the Kolyma River, near Cherskiy, Sakha
Republic, Russia. We found that the majority of C stored in the top 1 m of the watershed was
stored belowground (92%), with 19% in the top 10 cm of soil and 40% in the top 30 cm.  Carbon
was more variable in surface soils (10 cm; coefficient of variation (CV) = 0.35 between stands)
than in the top 30 cm (CV=0.14) or soil profile to 1 m (CV=0.20).  Combined active layer and
deep frozen deposits (surface - 15 m) contained 205 kg C m$^{-2}$ (yedoma, non-ice wedge) and 331
kg C m$^{-2}$ (alas), which, even when accounting for landscape-level ice content, is an order of
magnitude more C than that stored in the top meter of soil and two orders of magnitude more C
than in aboveground biomass. Aboveground biomass was composed of primarily larch (53%) but
also included understory vegetation (30%), woody debris (11%) and snag (6%) biomass.  While
aboveground biomass contained relatively little (8%) of the C stocks in the watershed,
aboveground processes were linked to thaw depth and belowground C storage.  Thaw depth was
negatively related to stand age, and soil C density (top 10 cm) was positively related to soil
moisture and negatively related to moss and lichen cover.  These results suggest that as the
climate warms, changes in stand age and structure may be as important as direct climate effects
on belowground environmental conditions and permafrost C vulnerability.

## 1  INTRODUCTION

Boreal forests cover roughly 22% of the earth's terrestrial landscape (Chapin et al., 2000) and account for approximately 9% of the global vegetation carbon (C) stock (Carvalhais et al., 2014).  Most of the C in boreal forests, however, is stored in the soil (Pan et al., 2011), where cold and wet conditions have limited microbial decomposition, and as a result, C has accumulated over the past several millennia (Hobbie et al., 2000; Trumbore and Harden, 1997). Recent estimates suggest that continuous and discontinuous permafrost in the boreal region store around 137 Pg, or 40% of near surface permafrost (< 1 m) C (Loranty et al., 2016).  Despite the massive amount of C present in the boreal region, the quantity of C stored here and the magnitude of the change in C stocks that will result from climate change is one of the least understood carbon-climate feedbacks (Schuur et al., 2015).

Over the past fifty years, air temperatures in the Arctic have risen nearly twice the global average as a result of climate change (Christensen et al., 2013), and this accelerated rate of warming means that the vast amount of C stored in high latitude systems is vulnerable to loss to the atmosphere (Koven et al., 2015; Schuur et al., 2015). The amount of C released as a result of thaw will be highly dependent on concurrent changes in topography and hydrology (Liljedahl et al., 2016; Schneider Von Deimling et al., 2015), vegetation (Guay et al., 2014; Sturm et al., 2005) fire regimes (Berner et al., 2012; Kasischke and Turetsky, 2006; Rogers et al., 2015; Soja et al., 2007), nutrient availability (Mack et al., 2004; Salmon et al., 2016), and as soil organic C lability (Harden et al., 2012; Schädel et al., 2014). Yet despite the vulnerability of permafrost soils to increased thaw and C release due to climate change, there is a lack of data quantifying the C stocks in northern latitudes compared to other regions.

Permafrost C pool estimates tend to be dominated by sites located in Alaska or western
Russia, with very few data points from the Russian low Arctic or Canadian high Arctic (Hugelius
et al., 2014; Tarnocai et al., 2009).  As a result, many regions are under-represented in
circumpolar permafrost C estimates (Hugelius et al., 2014; Johnson et al., 2011; Mishra et al.,
2013; Tarnocai et al., 2009).  Even in Alaska, which is one of the most densely sampled Arctic
sub-regions, Mishra and Riley (2012) found that the current sample distribution is insufficient to
characterize regional soil organic C (SOC) stocks fully because of SOC variation across
vegetation types, topography, and parent material. Furthermore, permafrost regions are
characterized by high heterogeneity in soil C stocks due to variability in soil-forming factors
(Vitharana et al., 2017) and at small spatial scales due to cryogenic processes (i.e., cryoturbation
at the sub-meter scale).  As a result, sampling at higher spatial resolution may provide more
accurate estimates of soil C stocks (Johnson et al., 2011; Tarnocai et al., 2009). Therefore,
understanding variation in soil properties at the meter scale is critical for reducing uncertainty in
estimates of current and future permafrost C pools (Beer, 2016).
Pleistocene-aged, C and ice rich permafrost (i.e. yedoma) deposits occur across Siberia
and Alaska (Strauss et al., 2013) and are particularly important for regional soil C estimates.
Yedoma deposits froze relatively quickly in geologic history (Schirrmeister et al., 2011; Zimov
et al., 2006), and as a consequence, these deep deposits (on average 25 m; Zimov et al. 2006) are
C rich compared to some other permafrost soils (Strauss et al., 2013; Zimov et al., 2006).
Approximately 30% of high latitude permafrost C is found in these yedoma deposits, even
though they comprise only 7% of the landscape (Walter Anthony et al., 2014).  However, due to
limited sampling of deep (> 3 m) permafrost, establishing how much C is in these deposits is
difficult, leading to high uncertainty in estimates of soil C pools in yedoma deposits (Strauss et
al., 2013; Walter Anthony et al., 2014).
While vegetation stores a relatively small portion of the C pool in boreal forests
(approximately 20%; Pan et al., 2011), it plays a crucial role in local and global C cycling, and
many future changes in C fluxes in this biome will likely occur as a result of changes in
vegetation (Elmendorf et al., 2012; Euskirchen et al., 2009; Myers-Smith et al., 2015; Swann et
al., 2010).  With increased temperatures, boreal forests are susceptible to insect invasions (Berg
et al., 2006; Kurz et al., 2008), moisture stress (Beck et al., 2011; Trahan and Schubert, 2016;
Walker et al., 2015), tree line advance and retrogression (Lloyd, 2005; Pearson et al., 2013), and
more frequent forest fires (Kasischke and Turetsky, 2006; Rogers et al., 2015; Soja et al., 2007),
which all have the potential to alter C cycling significantly in the region.  Importantly, climate-
change driven alterations in forest cover, composition, and structure will influence regional
energy balance through impacts on surface albedo, evapotranspiration, and ground insulation,
which will in turn affect ground thaw and soil C cycling (Chapin et al., 2005; Euskirchen et al.,
2009; Fisher et al., 2016; Jean and Payette, 2014; Loranty et al., 2014).
However, the aboveground processes that regulate C dynamics are not homogenous
throughout the boreal biome (Goetz et al., 2007).  For example, the fire regimes of larch (*Larix*
*spp.*) and pine (*Pinus sylvestris*) forests in Siberia are typically dominated by low to medium
intensity fires whereas dark coniferous forests common in Alaska and Canada are characterized
by higher intensity and severity fires (Rogers et al., 2015; Soja et al., 2006, 2007; Tautenhahn et
al., 2016).  The dynamics of larch forests are particularly important, as they store more than
twice the amount of SOC of all other boreal forest types in the continuous permafrost zone
combined (Loranty et al., 2016).  Despite this, larch forests in Siberia are notably under studied;
indeed, the estimate of C stored in Russian forests is the least well constrained of all forest
systems globally (Shuman et al., 2013).

In this study, we aim to reduce the uncertainty of regional C estimates by providing a

comprehensive assessment of vegetation, active layer, and permafrost C stocks in the Kolyma
River watershed in Northeast Siberia, Russia.  We present aboveground and belowground (to 1
m) C stocks from data collected from 20 sites across the watershed along with deep permafrost C
pools to 15 m depth from a yedoma deposit and an alas (thermokarst depression). We compare
variation in soil C pools at meter to kilometer scales in order to quantify the variability of
permafrost C at small spatial scales.  Additionally, we examine the drivers of thaw depth and C
density of active layer soils to understand environmental controls over these variables across the
watershed.  Together, these analyses allow us to estimate C pools and controls over changes in
these pools that will likely occur with climate change.

**2   METHODS**
**2.1   Site description**

Our study area was a watershed ('Y4 watershed', ~3 km$^2$; Figure 1) located within the

Kolyma River basin, which is the largest river basin (650,000 km$^2$) completely underlain by
continuous permafrost (Holmes et al., 2012).  The Y4 watershed is located near Cherskiy, Sakha
Republic, Russia approximately 130 km south of the Arctic Ocean and is underlain by yedoma,
which is widespread across the region (Grosse et al., 2013). The climate is continental with short,
warm summers (Jul avg: 12 °C) and long, cold winters (Jan avg: -33 °C). Annual precipitation is
low (~230 mm) and often occurs during summer (Cherskiy Meteorological Station; S. Davydov,
unpub data). Mean summer temperatures in this region increased by 1°C from 1938 to 2009
(Berner et al., 2013).

There are two main types of cryogenic deposits within the watershed. Upland areas are

Late Pleistocene syngenetic ice rich deposits of yedoma.  Drained thaw lake depressions are
underlain by alas consisting of lacustrine-wetland sediments in the upper pedon and taberal (i.e.
yedoma that thawed in a talik) deposits in the lower part of the profile. Permafrost temperatures
at 15 m vary from -2.8°C at the hilltops with relatively thin organic layers to -4°C in thermokarst
depressions with thick (up to 20 cm) moss and peat layers (A. Kholodov, unpub data).

Forests in the watershed are composed of a single larch species, *Larix cajanderi*, with a

well-developed understory of deciduous shrubs (primarily *Betula nana*, *Salix* spp., and
*Vaccinium uliginosum*), evergreen shrubs (e.g. *Vaccinium vitis-idaea, Empetrum nigrum,*
*Rhododendron subarcticum*), forbs (e.g., *Equisetum scirpoides, Pyrola* spp., and *Valeriana*
*capitate*), graminoids (*Calamagrostis* spp.), moss (e.g. *Aulacomnium palustre, Dicranum* spp.,
and *Polytrichum* spp ), and lichen (e.g. *Cladonia* spp, *Peltigera aphthosa,* and *Flavocetraria*
*cucullata*).

**2.2   Site selection and sampling design**

We selected 20 stands (i.e. 'sites') in the Y4 watershed that spanned a range of tree

aboveground biomass, as inferred from tree shadows mapped using high-resolution (50 cm)
WorldView-1 satellite imagery (Berner et al., 2012; Figure 1).  All sites were located in forested
stands except for one in a *Salix*-dominated riparian zone (Site 17) and another in a *Sphagnum*-
dominated alas (Site 18; Table 1).  Within each site, we established three 20 m long by 2 m wide
plots, each of which was separated by 8 m and ran parallel to slope contours (Figure S1).  In the
absence of a discernable slope, transects were aligned north-south. All sampling was conducted
in July 2012 and 2013 except stand age, which was sampled in 2016.

**2.3   Stand Age**

To determine stand age, we collected a wood slab or core from the base (~ 30 cm above

the organic layer) of 5-10 trees sampled randomly within each stand. Wood samples were dried
at 60 °C and then sanded sequentially with finer grit sizes to obtain a smooth surface. Each
sample was then scanned and the annual growth rings were counted using WinDendro (Regent
Instruments, Inc., Ontario).

**2.4   Solar Insolation and Slope**

Slope and aspect at each site were determined from a 4-m-resolution digital elevation

model of the watershed created by the Polar Geospatial Center (http://www.pgc.umn.edu/) using
stereo-pairs of World ViewX imagery.  Solar insolation was estimated using the Solar Radiation
analyses toolset in ArcGIS version 10 (ESRI , Redlands, CA). The toolset used variability in the
orientation (slope and aspect) to calculate direct and diffuse radiation for each pixel of the
elevation model in the Y4 watershed using viewshed algorithms (Fu and Rich, 2002; Rich et al.,
1994).  We report total insolation on the summer solstice for each pixel.

**2.5   Aboveground biomass**

We measured diameter at breast height (DBH; 1.4 m height) or basal diameter (BD; < 1.4

m height) of all trees and snags (i.e., dead trees standing $\geq 45°$ to the forest floor) within each 40-
$m^2$ plot (n= 3/site). Live and dead aboveground tree biomass were determined based on
allometric equations developed from *L. cajanderi* trees harvested near Cherskiy (Alexander et
al., 2012). Tree biomass was converted to C mass using a C concentration of 46% C for foliage
(live trees only), 47% C for stemwood/bark and snag, and 48% C for branches (Alexander et al.,

2012).

We estimated understory percent cover in six 1-m$^2$ subplots at each site; subplots were
placed at both ends of each of the three plots (at 0 and 20 m; Figure S1).  Understory vegetation
was sorted into functional types, which included shrub (evergreen and deciduous), herbs (forb
and graminoids), moss, lichen, and other (litter, woody debris and bare ground). In each site,
understory vascular plant biomass was determined in three 0.25 m$^2$ quadrats, each of which was
located within one of the percent cover plots.  We measured basal diameter of tall deciduous
shrubs (*Alnus* spp., *B. nana*, and *Salix* spp.) and used published allometric relationships to derive
biomass (Berner et al., 2015).  All remaining vascular plants were harvested and dried at 60 ºC
for 48 hours for dry mass determination.  We converted live understory biomass values to C
pools by multiplying biomass by 48% C content.
Following the line-intercept method for measuring woody debris (Brown, 1974), we set a
20-m transect along the middle of each plot, and counted the number of times woody debris
intercepted the transect for Class I fine woody debris (FWD; 0.0-0.49 cm diameter) and Class II
FWD (0.5-0.99 cm) along the first 2 m, Class III FWD (1.0 – 2.99cm) along the first 10 m, and
classes IV FWD (3.0-4.99 cm), V FWD (5.0-6.99 cm), and downed coarse woody debris (CWD;
> 7 cm diameter) along the entire 20 m length. We calculated the mass of woody debris
according to Alexander et al. (2012) using previously published multipliers for softwood boreal
trees from the Northwest Territories of Canada for FWD (Nalder et al., 1997) and decay class
and density values for softwood boreal tree species within Ontario, Canada for CWD (Ter-
Mikaelian et al., 2008). Mass values were converted to C pools based on average C
concentration of *L. cajanderi* boles (47% C). Total aboveground biomass (AGB) is reported as
the sum of the C pools in woody debris, snags, trees, and understory biomass.

**2.6 Canopy cover and leaf area index**
We measured canopy cover under uniform, diffuse light conditions at the center of each
site in four cardinal directions using a convex spherical densitometer, and Leaf Area Index (LAI)
using both hemispherical photography and an LAI-2000 Plant Canopy Analyzer (Li-COR,
Nebraska, NE, USA). The LAI-2000 was placed ~1 m above the ground at the center of each
site, and LAI estimates were divided by a factor of 0.68 (Chen et al., 2005) to account for foliage
clumping (Chen et al., 1997). Hemispherical photographs were taken ~1 m off the ground using
a Sigma SD 15 digital reflex camera with Sigma 4.5 mm F2.8 EX DC circular fisheye lens. A
N-S reflector was used for N orientation, and photographs were taken using automatic settings at
the center of each of the three transects at each site. The hemispherical photographs were
analyzed using Hemiview software.

**2.7 Thaw depth/organic layer depth**
We measured thaw depth using a metal thaw probe every meter along a 20 m transect
placed along the center of each plot (measured from 9 July through 3 August; does not represent
maximum thaw). Organic layer depth (OLD) was measured at 5 m intervals along each transect
by cutting through the active layer soil with a serrated knife and visually identifying and
measuring the depth to the organic-mineral boundary.

**2.8  Soil sampling and analysis**

Active layer soils were collected from all sites.  Surface permafrost soils (approximately

the top 60 cm of frozen soil, which contained some frozen active layer soil) were sampled at
seven sites (3 cores per site), and deep permafrost (15 m depth) was sampled at two sites (Sites
18 and 19).  We collected six active layer samples from each site, one at each end of the 20-m-
long plots.  We used a serrated knife to collect an 8 cm x 8 cm sample from the organic layer,
and a 2 cm diameter manual corer to collect the top 10 cm of mineral soil. When less than 5 cm
of mineral soil was thawed at the time of sampling, the mineral soil sample was excluded from
analysis (n=5).  At the seven sites where surface permafrost was sampled, we collected mineral
soil to frozen ground (average 28 cm thawed mineral soil depth) using a manual corer, and
sampled approximately 60 cm depth of frozen soil with a Soil Ice and Permafrost Research
Experiment (SIPRE) auger (7.62 cm diameter).  We collected two deep permafrost cores with a
rotary drill rig (UKB-12/25, Drilling Technology Factory); one deep core was collected from a
site underlain by yedoma and the other from an alas. Carbon pools presented for deep permafrost
include C in the active layer sampled at the drilling location.  Carbon pools reported for 1 m
depth were calculated using the seven surface permafrost samples as well as the top 1 m of the
deep core from the yedoma site.  All permafrost samples were kept frozen until analyzed as
described below.

Surface permafrost cores were sectioned into 10 cm increments. Coarse-roots (> 2 mm)

were removed from all active layer and surface permafrost soils, and fine roots and organic soils
were dried at 60 °C for 48 hours while mineral soils were dried at 105 °C for at least 48 hours.
Gravimetric water content (GWC) was determined as the ratio of soil water mass to soil dry
mass, and was reported as a percentage (i.e., GWC x 100).  Organic matter content was
measured as the percent mass lost from dried soil after combusting for 4 hours at 450°C.  Soil C
content was analyzed on a subset of soils (35 of 111 organic soils; 119 of 271 active layer and
surface permafrost mineral soil; and 30 of 149 deep permafrost samples) on a Costech CHN
analyzer at St. Olaf College or at the University of Georgia Stable Isotope Ecology Lab.  Carbon
concentrations of the full set of soil samples were then modeled using a linear relationship
between organic matter content and %C (C% = 0.524 * OM% – 0.575; $R^2$=0.96 for active layer
and surface permafrost; C% = 0.391 * OM% – 0.103; $R^2$=0.86 for deep permafrost samples).
Carbon content of coarse roots was assumed to be 50%.  Sampled soils were reclassified as
organic or mineral as needed (< 1% of samples) based on soil C content (C ≥ 20% for organic
soils).
Bulk density (BD) was determined as the mass of dry soil per unit volume (g cm$^{-3}$).
Volume of active layer soil samples was determined by measuring the ground area and depth
from where the soil sample was removed.  Volume of permafrost samples was quantified by
water displacement.  Ice volume was determined based on soil water content and assuming an ice
density of 0.9167 g cm$^{-3}$.
Soil C stocks in each depth increment were calculated as the product of %C, BD and soil
depth.  For the deep permafrost samples, sub-samples used for %C, %OM, and BD
measurements were collected from adjacent depth increments; therefore, for the %C-%OM
regression and C pool calculations, we used adjacent depth increments or interpolated values
between two adjacent depths.

**2.9   Statistical analysis**
To compare the variance in soil C among sites and studies, we used the coefficient of
variation (CV), which is the ratio of the standard deviation to the mean.  The CV is independent
of the unit or magnitude and can be used to compare intra-site variation (how variable the data
are relative to the mean value) among sites even if the mean of the sites is vastly different.  We
also used percent variation, which was calculated by subtracting the minimum value from the
maximum value and dividing by the maximum value.
We used a linear model to determine the relationship between canopy cover and LAI and
larch biomass and the relationship between the different components of AGB. To determine
potential environmental drivers of thaw depth and soil C, we fit a mixed effects linear model
using the nlme package in R (Pinherio et al., 2013), using average plot-level data (3/site) as a
replicate for each site.  The fixed effects were the environmental variables, and the random effect
was the nested study design (plots within sites).  Both thaw depth and soil C were log-
transformed to meet the assumption of normality.  After collinear explanatory variables were
removed from analysis using a variance inflation factor of three (as suggested by Zuur et al.
(2009)), we considered densiometry, organic layer depth, stand age, live shrub biomass, woody
debris, tree density, snag density, summer insolation, percent herbaceous cover, percent moss
cover, percent lichen cover, percent other cover, soil C, BD, and root C, as explanatory variables
for the thaw depth model.  For the soil C model the environmental variables considered were:
slope, summer insolation, snag biomass, live tree biomass, live shrub biomass, woody debris,
tree density, percent herbaceous cover, percent moss cover, percent lichen cover, percent other
cover, thaw depth, organic layer depth, root carbon, and moisture.  The best model for each
analysis was selected using backwards stepwise reduction of variables to obtain the lowest
*Akaike information criterion* (AIC) and the residuals of all final models were checked for
normality and homogeneity of variance (Burnham and Anderson, 2002).
All reported errors are the standard error of the mean.  All statistical analyses were
conducted using the statistical program R (R Core Development Team, 2012).

**3 RESULTS**
**3.1 Distribution of carbon pools**
The majority of C in the watershed to 1 m depth was stored belowground (92%; $10.9 \pm$
$0.8$ kg C m$^{-2}$ in top 1 m; Figure 2), with 19% in the top 10 cm of soil and 40% in the top 30 cm.
The top 10 cm of soil alone contained 58% more C than the total aboveground C stocks.

**3.2 Stand density, stand age, and aboveground biomass**
Stand density in the watershed ranged from 0.01 to 0.43 trees m$^{-2}$ in the forested sites
(mean density was $0.07 \pm 0.02$ trees m$^{-2}$; Table 2). Mean stand age was 150 ($\pm17$) yrs (Table 1),
but there was a large range in tree ages among sites (23-221 yrs) and within sites (average range:
78 yrs; maximum range: 238 yrs; minimum range: 7 yrs; Table S1).
Total C in AGB averaged $959 \pm 150$ g C m$^{-2}$ across sites in the watershed, with 53% in
larch biomass ($460 \pm 77$ g C m$^{-2}$), 30% in understory biomass ($254 \pm 28$ g C m$^{-2}$) 11% in woody
debris ($94 \pm 16.5$ g C m$^{-2}$), and 6% in standing dead tree mass ($55 \pm 19$ g C m$^{-2}$) (Figure 2; Table
3). Among sites across the watershed, aboveground C varied up to 95%. Together, all C in AGB
contributed 8% to the total amount of C stored above and belowground (to 1 m) across the
watershed.  Mean stand age was positively related to mean stand AGB $R^2=0.21$, $p<0.001$ and
negatively related to mean stand thaw depth ($R^2=0.58$, $p<0.001$).
Larch aboveground biomass was also highly variable across the watershed, with some
sites as low as 0 or 1.7 g C m$^{-2}$ and others as high as 1,340 and 1,362 g C m$^{-2}$. Of the three
techniques used for estimating canopy cover, LAI values from hemispherical photography (Table
2) showed the highest correlation with larch biomass ($R^2$= 0.69, p <0.001), but larch biomass
was also significantly associated with canopy density ($R^2$= 0.5, p< 0.001). There was no
relationship between larch biomass and understory biomass (p = 0.4); however, the percent cover
of tall shrubs was negatively related to both moss ($R^2$=0.2, p<0.001) and lichen cover ($R^2$=0.2,
p<0.001).
**3.3   Surface soils**
Average C content of the organic horizon was 37.6 (± 0.8) %C, whereas C content of the
thawed mineral horizon (0-10 cm) was 4.6 (± 0.48) %C. There were 2.24 (± 1.22) kg C m$^{-2}$
stored in the organic layer (average organic layer depth=11.2 ± 0.2 cm) and 1.96 (± 0.07) kg C
m$^{-2}$ in the top 10 cm of the mineral layer (Table 4).
There was large variation in BD, soil moisture (GWC), soil C content and thaw depth
among sites (Table 5). Carbon content and GWC were more variable in mineral soils than in
organic (CV$_{mineral}$ = 0.55 for %C and 0.48 for GWC; CV$_{organic}$ = 0.15 for %C and 0.36 for GWC),
while BD was more variable in organic soils (CV$_{organic}$ = 0.51; CV$_{mineral}$ = 0.3). While the CV of
thaw depth was not particularly high (0.28), the difference between the sites with the highest and
lowest thaw depth measured was still 65%, underscoring the heterogeneity of soil properties
across the watershed. Variation in thaw depth was primarily due to stand age (Figure 3; Table
S2).
Soil C density in the top 10 cm of the ground surface (i.e., 0-10 cm soil depth, which may
have contained both organic and mineral soils) varied up to 93% across the watershed (range:
0.51-7.14 kg C m$^{-2}$; Table 4; Table S2), but the coefficient of variation (CV) was larger within
sites (0.32) than it was between sites (0.26), indicating that soil C is more variable at the meter
scale than it is at the kilometer scale. The distribution of soil C density in the top 10 cm was best
explained by soil moisture, percent moss, and percent lichen cover (Table S2); soil C density was
positively related to soil moisture and negatively related to percent moss and lichen cover
(Figure 4).
Soil in the top 30 cm of the profile contained on average $4.8 \pm 0.3$ kg C m$^{-2}$, but soil C
density in the top 30 cm varied by 56% across the watershed as a whole. The average CV within
a site was 0.16 whereas the CV among sites was 0.22, indicating C density at 30 cm is similar or
more variable across the watershed than at the meter scale. The top 1 m of soil contained $10.9 \pm$
0.8 kg C m$^{-2}$($13.8 \pm 3.0$ kg C m$^{-2}$ with alas site; Table S4). Soil C in the top 1 m varied by 63%
across the watershed and by 44% among sites. The average CV within a site was 0.15 whereas
among sites the CV was 0.20, indicating soil C to 1 m is similarly variable at the meter and
kilometer scales. Ice content in the top 1 m was on average $68 \pm 2\%$ by volume, with a range of
between 51% and 80%.
**3.5   Deep permafrost soils**
Deep permafrost soils (includes surface active layer to 15 m) contained 205 kg C m$^{-2}$ (site
19; yedoma deposit, non-ice wedge) and 331 kg C m$^{-2}$ (site 18; alas). Carbon density at each 1
m interval ranged from 7.87-21.63 kg C m$^{-3}$ in the yedoma deposit and 6.9-14.5 kg C m$^{-3}$ in the
deeper portion of the alas (Figure 5; Table S5). The top 2 m of the alas were characterized by
particularly high C density (~30 kg m$^{-3}$).
Highlighting the variability of C in deep permafrost, the total soil C density in the two
cores varied by 38%. The alas site had higher GWC than the yedoma site in the first 2 m (GWC:
$385 \pm 81\%$ and $41 \pm 8$ %, respectively).  Throughout the entire profile, GWC was $46 \pm 2\%$ in the
yedoma core and $100 \pm 23\%$ in the alas core.  Overall, BD was similar between the two cores,
and most of the variation in BD occurred in the top 5 m (Figure 5).

**4    DISCUSSION**
**4.1    Aboveground biomass**

Aboveground C pools within the Y4 watershed represented only a small fraction (8%) of

total C pools, likely due to low tree density at most sites ($< 0.09$ trees $m^{-2}$ in all but one site)
and/or young stand ages at a few sites. Low-density, mature ($> 75$ years old) stands with no
recent fire activity are common in this region (Berner et al. 2012); however, wildfires can
produce stands of considerably higher density ($> 3$ trees $m^{-2}$), which can substantially increase
AGB and contribution to total C pools as stands mature (Alexander et al. 2012).  Aboveground C
pools were similar to those reported by Alexander et al. (2012) for 17 nearby stands of similar
age and density, but C in larch AGB was lower ($\sim 23\%$) than a landscape-level estimate ($\sim 600$ g
C $m^{-2}$) across the Kolyma River basin (Berner et al. 2012). Our estimate for C stored in larch
AGB was also four times lower than that of a mature (155-yr old), mid-density (0.19 trees $m^{-2}$)
stand near Cherskiy and two times lower than a mature, low-density (0.08 trees $m^{-2}$) stand near
Oymyakon, south of Cherskiy (Kajimoto et al., 2006).  In addition, our larch AGB estimates fell
within the low range of larch stands across other high-latitude ($> 64°$ N) regions and were
generally 3-10 times lower than other stands (Kajimoto et al., 2010).  Our considerably lower
estimates reflect both the sparse, open grown structure of our stands (Osawa and Kajimoto,
2010) and the poor soil environment (e.g., shallow rooting zone, low soil temperature, low N
availability) found in stands near latitudinal and altitudinal treeline (Kajimoto et al. 2010).
Despite the small contribution of AGB to total C pools across our stands, aboveground
vegetation composition and structure were important factors related to soil C pools and
permafrost thaw (see below). In addition, characteristics of aboveground vegetation are major
determinants of land-atmosphere C fluxes (Bradshaw and Warkentin, 2015) and thus remain
essential components of C dynamics even when pools are relatively low.
**4.2   Variability of soil C pools**
Soil C density is controlled by numerous biogeophysical factors such as climate, local
geomorphology, soil parent material, time since last disturbance, and vegetation type, all of
which lead to high variability in soil C pools at the regional and local scale.  Our soil C pool
estimates for a Siberian larch forest watershed fall within the range of published assessments that
characterize this area (Alexander et al. 2012; Broderick et al. 2015), but are at the low end of
other studies (Alexeyev and Birdsey, 1998; Hugelius et al., 2014; Matsuura et al., 2005; Palmtag
et al., 2015; Stolbovoi, 2006).  For example, our mean estimate of $4.8 \pm 1$ kg C m$^{-2}$ in the top 30
cm of soil is less than half of a published assessment of C stored in soils across Russian larch
forests (10.2 kg C m$^{-2}$; Stolbovoi, 2006), and less than one third of the mean estimate for Turbel
soils across the permafrost region (14.7 kg C m$^{-2}$; Hugelius et al., 2014); however, variation in
the permafrost region Turbel soil C pool is high (CV = 0.85; Hugelius et al., 2014), and our
mean estimate falls within one standard deviation of this regional mean.
Within larch forests, there is substantial variation in soil C pools at regional scales, driven
by variation in soil parent material and climate.  For example, larch forests in Northeastern
Siberia store significantly more C (16 kg C m$^{-2}$) in the active layer and have more variable soil C
pool estimates than larch forests in Central Siberia (6.3 kg C m$^{-2}$) (Matsuura and Hirobe, 2010).
There is also considerable variation in soil C pools within larch forests at smaller spatial scales.
Indeed, the active layer in larch forests located within 50 km from our study site contained twice
as much C as found in our study ($4.8 \pm 0.3$ kg C m$^{-2}$ to 30 cm); there was 8.3 kg C m$^{-2}$ in the
active layer (38 cm) of a larch forest 44 km from the Y4 watershed (Matsuura et al., 2005) and
$9.5 \pm 2.9$ (SD) kg C m$^{-2}$ in the top 30 cm of soils from a forest 3 km away (Palmtag et al., 2015).
This variation in soil C pools points to the extreme variability in soil C throughout the landscape,
even at the kilometer scale.  It also highlights the importance of sampling replication at small
scales; with 21 total soil cores at seven sites, our CV (0.13) was less than half of other studies
with lower site-level replication (Palmtag et al., 2015).

As the climate warms, C in surface permafrost is becoming increasingly vulnerable to

thawing and subsequent decomposition and loss to the atmosphere.  As such, estimating
variation in C pool size is critical for understanding permafrost climate feedbacks.  The C stored
in the top 1 m of Y4 soils ($10.9 \pm 0.8$ kg C m$^{-2}$) was similar to the average 1-m C pool reported
for the Yakutia region, which comprises a range of ecosystem types (8.1 kg C m$^{-2}$; Alexeyev and
Birdsey, 1998) but 37% lower than the 1 m soil C pool reported in a forest only 3 km away (17.3
$\pm 5.7$ kg C m$^{-2}$; Palmtag et al., 2015).  However, the percent difference between our estimate and
the nearby study (37%) was similar to the percent difference found between sites in the Y4
watershed (44%; Table 4), suggesting that these differences among studies are likely due to
natural variation in the landscape.

Carbon pool estimates from deep permafrost (>3 m) are limited across the Arctic

(Hugelius et al., 2014; Schuur et al., 2015; Tarnocai et al., 2009), yet these data are critical for
assessing variation in and controls on C density of yedoma, as these soils have particularly high
C density at depth (Strauss et al., 2013; Zimov et al., 2006).  The average carbon density of deep
permafrost from yedoma deposits in the Y4 watershed (13.5 kg C m$^{-3}$) was similar to values
reported for yedoma in pan-Arctic summary studies (10 +7/-6 kg C m$^{-3}$, Strauss et al. (2013);
13.0 ± 0.75 kg C m$^{-3}$ after correction for ice volume, Walter Anthony et al. (2014)) and in taiga
sites within 100 km of Cherskiy (12.3-15.4 kg C m$^{-3}$ after correction for ice volume, Walter
Anthony et al. (2014) and references therein; 14.3 kg C m$^{-3}$, Shmelev et al., 2017).  Carbon
density was almost twice as high in the alas, which is consistent with findings indicating that alas
and thermokarst soils store substantially more C (~ 40-70%; Walter Anthony et al. (2014);
Strauss et al. (2013); Siewert et al. (2010)) than undisturbed yedoma, a difference that is likely
due to higher rates of recent (Holocene) C accumulation at the alas site (Walter Anthony et al.,
2014).  Yedoma is characterized by high landscape-level ice content due to the prevalence of
large ice wedges, which can comprise 31 to 63% of ground volume (Ulrich et al., 2014).
Accounting for these deep ice deposits, which were not sampled in this study, would reduce our
landscape-level estimate of C content in the top 15 m of yedoma from 205 kg C m$^{-2}$ to 76-141 kg
C m$^{-2}$, which is still an order of magnitude more C than is stored in the active layer and two
orders of magnitude more C than is stored in biomass.
**4.3  Micro-scale variation in soil carbon and thaw depth**

In addition to the effects of parent material and climate on soil C storage, soil carbon

pools are determined by the balance between biological inputs and losses due to microbial
decomposition and lateral transport.  These biological processes are, in turn, also heavily
influenced by climate on regional and local scales.  We found that soil samples with higher
moisture content also had higher C density, which is likely due to both the effects of soil
moisture on microbial activity and indirect effects of soil moisture on C inputs to soils through
effects on plant productivity.  In wetter soils, oxygen diffusion is limited, resulting in anaerobic
conditions where microbial decomposition is slower, and C can accumulate at a higher rate than
in more well-drained, well-aerated soils (Schädel et al., 2016).  However, this positive
association between moisture and C density may also be a result of increased C inputs and plant
productivity associated with higher soil moisture (Berner et al. 2013) or the lateral movement of
dissolved organic C into the wetter sites.  It is likely that environmental controls on both C inputs
and losses are driving the patterns of C accumulation across the watershed.

Plant species composition may also play an important role in soil C storage in boreal

forests (Hollingsworth et al., 2008) through the quality and quantity of litter inputs and through
vegetation effects on environmental controls such as soil moisture and temperature.  Lichens and
mosses are sometimes thought to encourage soil C storage through their promotion of low soil
temperatures, higher moisture, and a relatively acidic environment (Bonan and Shugar, 1989).
However, at our sites, increasing abundance of lichen and moss was associated with lower soil C
storage, which may have been due to lower rates of C fixation (Turetsky et al., 2010), higher
rates of decomposition of vascular plant litter in moss and lichen patches (Wardle et al., 2003),
or impacts of vegetation functional types on soil moisture and soil temperatures. Because the
interactions between soil processes and vegetation are bidirectional, the processes driving these
observed patterns are unclear and further experimental work is needed to identify the
mechanisms.

Increasing thaw depth may result in increased C loss from boreal ecosystems; as more

soil is thawed, more organic matter is available for decomposition.  We found that thaw depth
was negatively related to stand age; the deeper thaw depth observed in the younger sites could be
a result of more recent burning events, which tend to increase thaw depth (O'Donnell et al.,
2011; Yoshikawa et al., 2002).

**5  CONCLUSIONS**

We found that the overwhelming majority of C in the Y4 watershed was stored

belowground, but that the amount of C within any given pool was highly variable throughout the
landscape; C storage in AGB varied up to 95% among sites, and there was 69% variation in the
top 10 cm of soil, 36% in the top 30 cm, and 28% in the top 1 m.   This variability among sites in
our study was similar to the variability between our sites and others that were 3 to 50 km away
(Matsuura et al., 2005; Palmtag et al., 2015), indicating a high level of natural variability at the
meter and kilometer scales.  Our results also indicate higher soil C variability in surface soils
when compared to deeper soils, indicating that recent, on-going processes significantly
contribute to soil C variability.  Specifically, our results show that soil moisture, aboveground
biomass, and vegetation community structure are influential in explaining near-surface
belowground C storage. These linkages between above and belowground processes, such as the
negative relationship between stand age and thaw depth, have important implications for soil C
vulnerability as tree lines shift and biomass and stand structure are increasingly impacted by fire,
climate, and direct human disturbances.

**DATA AVAILABILITY**

All data are available as supplemental material and through the Arctic Data Center

through the following citation: Kathryn Heard, Susan Natali, Andrew Bunn, and Heather D.
Alexander. 2015. Northeast Siberia Plant and Soil Data: Plant Composition and Cover, Plant and
Soil Carbon Pools, and Thaw Depth. NSF Arctic Data Center. doi:10.5065/D6NG4NP0.

**AUTHOR CONTRIBUTION**

E. Webb contributed to data collection and processing and analyzed data, created figures,

and drafted manuscript.  K. Heard collected, processed, and summarized data and contributed to
writing.  S. Natali oversaw and contributed to data collection, processing, analysis, and writing.
A. Bunn oversaw data collection, processing, and analysis. H. Alexander contributed to data
collection, analysis, and writing.  L. Berner contributed to data collection and processing and
figure creation.  M. Loranty contributed to data collection and processing.  J. Schade contributed
to lab analyses.  V. Spektor and A. Kholodov collected and processed deep permafrost cores. N.
Zimov contributed to data collection.  All authors reviewed the manuscript and provided critical
feedback.

**COMPETING INTERESTS**
The authors declare that they have no conflict of interest.

**ACKNOWLEDGMENTS**
This project was supported by funding from the National Geographic Society (Natali) and the
National Science Foundation (NSF-1044610, NSF-1417745 NSF-1014180, NSF-1044417, NSF-
1417700, NSF-1417908; Natali, Natali, Schade, Bunn, Loranty, Kholodov).  We thank S. Shin
and other Polaris Project 2013 participants for field and lab assistance, and the staff and scientists
at the Northeast Science Station for logistical and field support.

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

 **FIGURE DESCRIPTIONS**


**Figure 1.** Location of the Y4 watershed in relation to Russia (inset) and location of the sampling
sites within the Y4 catchment.

**Figure 2.** Average carbon density of all sites in the Y4 watershed (top: above and
belowground to 1 m; bottom: aboveground only). Bars indicate standard error.

**Figure 3.** Relationship between thaw depth and stand age. Each point represents the
average thaw depth measurement taken along a transect (three transects/site) and the stand age
of the entire site.  Thaw depths were measured in July/August of 2012 and 2013.

**Figure 4.** Relationship between SOC in the top 10 cm of soil and moisture, moss cover, and
lichen cover. Each point represents the average SOC measured at each transect (three
transects/site) and its corresponding moisture content or the average moss or lichen cover
measured at that transect.

**Figure 5.** Bulk density, carbon density, and ice content of the two deep (15 m) permafrost soil
cores.

 **TABLES**

**Table 1:** Site Characteristics. All sites were in forested areas except #17 (riparian); Site #18 (alas) had few scattered trees located along one end of the sampling transects.

| Site Number | Latitude (Degrees North) | Longitude (Degrees East) | Slope (Degrees) | Aspect (Degrees) | Summer Insolation (WH m$^{-2}$) | Stand Age (yrs) |
|---|---|---|---|---|---|---|
| 1 | 68.74747 | 161.38988 | 5 | 160 | 4507 | 155 |
| 2 | 68.74529 | 161.38908 | 10 | 8 | 3950 | 167 |
| 3 | 68.74472 | 161.41486 | 14 | 249 | 4399 | 203 |
| 4 | 68.74164 | 161.41562 | 9 | 245 | 4409 | 23 |
| 5 | 68.74834 | 161.41350 | 10 | 357 | 3954 | 218 |
| 6 | 68.74939 | 161.41759 | 8 | 225 | 4509 | 205 |
| 7 | 68.74915 | 161.39000 | 5 | 57 | 4239 | 155 |
| 8 | 68.74932 | 161.38820 | 7 | 36 | 4132 | 208 |
| 9 | 68.75267 | 161.38544 | 8 | 340 | 4038 | 202 |
| 10 | 68.75352 | 161.39455 | 16 | 72 | 4008 | 211 |
| 11 | 68.74869 | 161.40834 | 10 | 222 | 4533 | 123 |
| 12 | 68.74837 | 161.40237 | 10 | 63 | 4121 | 71 |
| 13 | 68.74660 | 161.40433 | 17 | 61 | 3856 | 179 |
| 14 | 68.74513 | 161.40063 | 1 | 103 | 4361 | 40 |
| 15 | 68.75188 | 161.39095 | 3 | 237 | 4410 | 221 |
| 16 | 68.75519 | 161.40013 | 3 | 294 | 4307 | 200 |
| 17 | 68.74152 | 161.41411 | 8 | 225 | 4479 | - |
| 18 | 68.74632 | 161.38776 | 3 | 84 | 4314 | - |
| 19 | 68.74479 | 161.38410 | 6 | 61 | 4231 | 26 |
| 20 | 68.74333 | 161.40688 | 5 | 124 | 4429 | - |


**Table 2:** Leaf area index (LAI), tree and snag density, and percent cover of the 20 plots in the Y4 watershed. Values in parenthesis are standard error of the mean. Other cover includes woody debris and bare ground.

| Site Number | LAI (Hemispherical Photography) | LAI (LAI-2000) | Larch Density (trees/m$^2$) | Snag Density (snags/m$^2$) | Canopy Cover (%) | Understory Shrub Cover (%) | Herbaceous cover (%) | Moss Cover (%) | Lichen Cover (%) | Other Cover (%) |
|---|---|---|---|---|---|---|---|---|---|---|
| 1 | 0.03 (0.00) | 0.13 | 0.09 (0.05) | 0.00 | 22.4 (3.2) | 45.2 (2.7) | 3.5 (1.7) | 22.0 (3.4) | 15.6 (4.9) | 12.4 (3.4) |
| 2 | 0.22 (0.02) | 0.13 | 0.04 (0.00) | 0.00 | 16.0 (4.0) | 49.4 (5.4) | 4.8 (2.4) | 25.0 (4.4) | 6.9 (2.9) | 13.8 (6.0) |
| 3 | 0.53 (0.03) | 0.68 | 0.08 (0.03) | 0.00 | 43.2 (7.4) | 60.3 (9.0) | 0.7 (0.3) | 31.3 (9.4) | 3.4 (2.6) | 4.3 (0.6) |
| 4 | 0.02 (0.01) | 0.00 | 0.08 (0.07) | 0.00 | 2.6 (2.6) | 72.3 (7.9) | 2.5 (1.6) | 7.4 (2.4) | 3.4 (2.1) | 14.3 (5.7) |
| 5 | 0.37 (0.05) | 1.35 | 0.08 (0.02) | 0.03 (0.01) | 32.3 (7.6) | 51.5 (4.9) | 4.2 (1.4) | 14.4 (2.9) | 16.9 (4.1) | 13.1 (2.4) |
| 6 | 0.38 (0.03) | 0.47 | 0.06 (0.01) | 0.03 (0.01) | 26.0 (4.6) | 57.9 (7.2) | 8.4 (5.9) | 17.4 (5.2) | 3.6 (1.3) | 12.1 (3.8) |
| 7 | 0.15 (0.08) | 0.00 | 0.05 (0.02) | 0.00 | 17.6 (8.4) | 34.8 (3.5) | 3.4 (0.8) | 34.0 (7.1) | 22.8 (6.4) | 4.8 (1.9) |
| 8 | 0.06 (0.04) | 0.29 | 0.02 (0.00) | 0.00 | 7.0 (2.1) | 34.8 (4.5) | 3.8 (1.8) | 32.5 (7.9) | 24.8 (9.5) | 4.0 (2.3) |
| 9 | 0.07 (0.02) | 0.00 | 0.01 (0.00) | 0.00 | 9.4 (1.6) | 44.2 (5.5) | 0.0 | 33.5 (5.0) | 16.7 (7.6) | 5.6 (1.6) |
| 10 | 0.30 (0.09) | 1.41 | 0.08 (0.04) | 0.04 (0.02) | 24.3 (6.2) | 49.2 (10.6) | 8.6 (2.9) | 29.8 (8.8) | 5.3 (1.4) | 7.1 (2.5) |
| 11 | 0.05 (0.03) | 0.22 | 0.02 (0.01) | 0.00 | 4.7 (1.5) | 33.6 (6.9) | 5.8 (3.0) | 15.3 (4.5) | 30.6 (8.0) | 15.0 (5.9) |
| 12 | 0.01 (0.00) | 0.00 | 0.02 (0.01) | 0.00 | 0.0 (0.0) | 47.1 (7.4) | 7.5 (4.0) | 20.2 (3.7) | 19.0 (5.3) | 6.9 (3.2) |
| 13 | 0.23 (0.07) | 0.82 | 0.07 (0.01) | 0.02 (0.01) | 18.9 (3.0) | 47.4 (8.1) | 4.2 (2.6) | 25.6 (8.2) | 13.6 (6.2) | 9.1 (0.8) |
| 14 | 0.00 (0.00) | 0.00 | 0.03 (0.02) | 0.00 | 0.8 (0.8) | 47.2 (12.0) | 5.8 (3.7) | 11.3 (3.8) | 33.5 (13.9) | 2.3 (1.1) |
| 15 | 0.03 (0.01) | 0.00 | 0.02 (0.01) | 0.00 | 3.8 (1.0) | 41.3 (3.9) | 3.8 (1.7) | 22.4 (4.5) | 21.9 (4.6) | 10.4 (5.5) |
| 16 | 0.31 (0.13) | 0.88 | 0.05 (0.01) | 0.00 | 18.5 (7.7) | 35.6 (7.6) | 2.2 (0.6) | 32.2 (11.6) | 25.9 (9.0) | 4.1 (1.5) |
| 17 | - | - | 0.0 | 0.00 | 13.9 (13.9) | 65.8 (15.1) | 11.1 (4.4) | 0.1 (0.1) | 0.1 (0.1) | 23.4 (11.5) |
| 18 | - | - | 0.01 (0.01) | 0.00 | 5.2 | 51.9 (6.5) | 12.5 (4.1) | 32.0 (5.0) | 0.2 (0.2) | 3.3 (1.9) |
| 19 | - | 2.03 | 0.43 (0.28) | 0.00 | 16.2 (2.2) | - | - | - | - | - |
| 20 | - | - | 0.06 (0.03) | 0.04 (0.02) | 6.1 (1.3) | - | - | - | - | - |

**Table 3:** Aboveground biomass (g C m$^{-2}$) at each site in the Y4 watershed. Total aboveground biomass is the sum of the larch, understory vascular, standing dead tree, and woody debris biomass. Understory vascular biomass does not include lichen and moss. Values in parenthesis are standard error of the mean.

| Site Number | Larch | Understory vascular | Shrub | Standing dead tree | Woody debris | Total live | Total dead | Total Aboveground |
|---|---|---|---|---|---|---|---|---|
| 1 | 392 (313) | 112 (41) | 52 (52) | 0 (0) | 322 (87) | 504 (304) | 322 (87) | 826 (389) |
| 2 | 603 (244) | 140 (50) | 75 (40) | 0 (0) | 76 (7) | 744 (213) | 76 (7) | 820 (217) |
| 3 | 743 (125) | 320 (106) | 209 (146) | 0 (0) | 86 (15) | 1063 (230) | 86 (15) | 1149 (235) |
| 4 | 67 (66) | 611 (166) | 529 (176) | 0 (0) | 59 (17) | 679 (153) | 59 (17) | 737 (167) |
| 5 | 1362 (516) | 193 (27) | 96 (32) | 219 (96) | 122 (28) | 1555 (490) | 341 (105) | 1896 (579) |
| 6 | 1340 (635) | 257 (81) | 146 (69) | 386 (236) | 131 (50) | 1597 (560) | 517 (218) | 2114 (361) |
| 7 | 263 (65) | 271 (86) | 209 (73) | 0 (0) | 24 (8) | 533 (45) | 24 (8) | 557 (52) |
| 8 | 471 (303) | 170 (115) | 124 (108) | 27 (27) | 10 (3) | 641 (294) | 37 (29) | 678 (319) |
| 9 | 122 (68) | 176 (93) | 64 (35) | 0 (0) | 37 (11) | 298 (60) | 37 (11) | 335 (65) |
| 10 | 697 (405) | 183 (64) | 51 (51) | 262 (140) | 106 (16) | 880 (400) | 368 (153) | 1248 (501) |
| 11 | 227 (201) | 185 (87) | 95 (95) | 0 (0) | 62 (17) | 413 (285) | 62 (17) | 475 (278) |
| 12 | 6 (6) | 116 (39) | 22 (13) | 0 (0) | 18 (4) | 122 (45) | 18 (4) | 140 (45) |
| 13 | 698 (124) | 139 (25) | 32 (18) | 93 (69) | 306 (189) | 837 (126) | 399 (146) | 1236 (217) |
| 14 | 5 (4) | 253 (184) | 169 (152) | 0 (0) | 16 (2) | 259 (183) | 16 (2) | 275 (181) |
| 15 | 142 (85) | 180 (41) | 82 (48) | 0 (0) | 71 (63) | 322 (59) | 71 (63) | 393 (6) |
| 16 | 984 (491) | 470 (256) | 417 (261) | 0 (0) | 56 (21) | 1454 (628) | 56 (21) | 1510 (633) |
| 17 | 0 (0) | 2657 (2575) | 2621 (2588) | 0 (0) | 118 (72) | 2657 (2575) | 118 (72) | 2775 (2642) |
| 18 | 2 (2) | 263 (46) | 245 (42) | 0 (0) | 16 (5) | 265 (47) | 16 (5) | 281 (50) |
| 19 | 35 (21) | 465 (172) | 382 (177) | 0 (0) | 116 (45) | 500 (159) | 116 (45) | 615 (196) |
| 20 | 585 (217) | 321 (163) | 156 (105) | 47 (26) | 158 (140) | 906 (173) | 205 (118) | 1111 (244) |


**Table 4:** Soil carbon in the Y4 watershed. Thawed soil cores were sampled from 6 locations per site. Permafrost cores were sampled to 1 m at 7 sites (3/site). Root C and soil C values were normalized to 10 cm. The combined soil C value is the amount of C in the top 10 cm of soil, regardless of soil type (mineral/organic). Carbon pools from the permafrost cores include active layer soil (0-30 or 0-100 cm from top of ground surface). Values in parenthesis are standard error of the mean.

| Site Number | Thawed Soil Cores | | | | | Permafrost Cores | |
|---|---|---|---|---|---|---|---|
| | Root C (g C m$^{-2}$) | | Soil C (kg C m$^{-2}$) | | | C in top 30 cm (kg C m$^{-3}$) | C in top 100 cm (kg C m$^{-3}$) |
| | Organic | Mineral | Organic | Mineral | Combined | | |
| 1 | 137 (27) | 0 | 2.60 (0.27) | 2.03 (0.21) | 2.34 (0.22) | 4.69 (0.06) | 9.36 (0.09) |
| 2 | 97 (60) | 0 | 1.35 (0.11) | 1.46 (0.32) | 1.32 (0.12) | 3.67 (0.34) | 10.16 (0.60) |
| 3 | 108 (42) | 0 | 1.86 (0.32) | 1.43 (0.19) | 1.83 (0.29) | | |
| 4 | 169 (183) | 0 | 2.06 (0.47) | 2.06 (0.22) | 2.49 (0.48) | | |
| 5 | 453 (108) | 0 | 4.47 (1.74) | 1.57 (0.05) | 3.42 (0.76) | | |
| 6 | 230 (169) | 0 | 3.86 (1.03) | 2.22 (0.43) | 3.71 (0.93) | | |
| 7 | 44 (22) | 0 | 1.13 (0.22) | 2.31 (0.41) | 1.14 (0.22) | 4.29 (0.32) | 10.48 (0.67) |
| 8 | 69 (25) | 0 | 1.25 (0.12) | 2.79 (0.67) | 1.38 (0.19) | | |
| 9 | 177 (17) | 45 (31) | 2.51 (0.26) | 1.54 (0.33) | 2.41 (0.40) | 4.85 (0.36) | 8.63 (0.71) |
| 10 | 278 (35) | 0 | 2.12 (0.45) | 1.36 (0.12) | 2.10 (0.46) | 4.82 (0.44) | 9.39 (0.06) |
| 11 | 520 (346) | 6 (4) | 1.63 (0.42) | 2.02 (0.16) | 1.66 (0.30) | | |
| 12 | 271 (87) | 0 | 1.39 (0.04) | 3.26 (0.83) | 1.51 (0.05) | | |
| 13 | 267 (30) | 0 | 1.65 (0.28) | 1.96 (0.29) | 1.66 (0.29) | | |
| 14 | 252 (74) | 6 (4) | 3.12 (0.47) | 1.31 (0.26) | 2.74 (0.15) | | |
| 15 | 103 (8) | 0 | 2.04 (0.58) | 2.15 (0.53) | 1.84 (0.38) | | |
| 16 | 189 (184) | 20 (11) | 1.70 (0.57) | 2.08 (0.49) | 1.66 (0.33) | 5.32 (1.19) | 11.90 (3.83) |
| 17 | 0 | 97 (35) | - | 2.37 (0.21) | 2.76 (0.78) | | |
| 18 | 95 (36) | 0 | 2.19 (0.40) | 2.66 (2.21) | 1.49 (0.55) | | |
| 19 | 205 (91) | 203 (152) | 3.51 (0.47) | 2.74 (1.23) | 2.85 (0.72) | | |
| 20 | 0 | 0 | 2.44 (0.70) | 1.41 (0.26) | 1.85 (0.43) | 5.70 (0.55) | 11.91 (0.90) |

**Table 5:** Properties of thawed soil in the Y4 watershed. The mineral layer was collected to approximately 10 cm below the organic layer (see methods). No relationship existed between sample date and thaw depth or sample date and water content. Values in parenthesis are standard error.

| Site Number | Thaw depth (cm) | Organic Layer Depth (cm) | Bulk Density (g cm$^{-3}$) | | Gravimetric Water Content (%) | | Carbon Content (%) | |
|---|---|---|---|---|---|---|---|---|
| | | | Organic | Mineral | Organic | Mineral | Organic | Mineral |
| 1 | 23 (1) | 13 (1) | 0.078 (0.021) | 0.52 (0.16) | 198.9 (34.4) | 64.7 (17.4) | 37.6 (3.5) | 6.9 (2.5) |
| 2 | 22 (1) | 11 (1) | 0.040 (0.011) | 0.64 (0.05) | 203.8 (28.0) | 33.9 (5.8) | 38.3 (4.1) | 2.4 (0.5) |
| 3 | 24 (1) | 14 (1) | 0.062 (0.011) | 0.70 (0.11) | 103.3 (16.2) | 29.1 (4.4) | 30.4 (2.2) | 2.3 (0.6) |
| 4 | 41 (2) | 10 (1) | 0.148 (0.063) | 0.54 (0.14) | 107.3 (28.9) | 61.0 (15.6) | 26.6 (4.0) | 8.7 (3.0) |
| 5 | 23 (1) | 8 (1) | 0.120 (0.032) | 1.02 (0.08) | 220.2 (23.1) | 25.6 (2.1) | 39.2 (3.2) | 1.6 (0.3) |
| 6 | 21 (2) | 9 (1) | 0.113 (0.039) | 0.63 (0.05) | 182.0 (19.8) | 34.2 (6.1) | 39.0 (3.0) | 3.8 (1.0) |
| 7 | 21 (1) | 12 (1) | 0.026 (0.005) | 0.76 (0.18) | 348.5 (48.4) | 43.6 (10.2) | 44.4 (2.0) | 3.9 (1.2) |
| 8 | 16 (1) | 11 (1) | 0.027 (0.002) | 0.68 (0.10) | 304.9 (32.1) | 46.4 (10.3) | 46.7 (0.6) | 4.4 (1.1) |
| 9 | 26 (2) | 13 (1) | 0.082 (0.010) | 0.64 (0.12) | 171.3 (29.5) | 46.5 (11.2) | 30.9 (4.4) | 5.5 (2.1) |
| 10 | 23 (1) | 11 (1) | 0.048 (0.007) | 0.89 (0.05) | 272.6 (15.2) | 26.5 (1.7) | 43.6 (1.9) | 1.6 (0.2) |
| 11 | 35 (2) | 10 (1) | 0.060 (0.023) | 0.84 (0.12) | 142.8 (17.8) | 39.4 (6.9) | 30.5 (3.3) | 3.6 (1.6) |
| 12 | 29 (2) | 10 (1) | 0.053 (0.020) | 0.67 (0.10) | 247.7 (17.5) | 58.3 (10.7) | 43.5 (1.8) | 5.0 (1.0) |
| 13 | 29 (1) | 12 (1) | 0.042 (0.008) | 0.71 (0.11) | 194.1 (15.4) | 48.6 (12.6) | 40.0 (1.4) | 4.0 (1.0) |
| 14 | 42 (2) | 8 (1) | 0.103 (0.016) | 0.82 (0.10) | 165.8 (14.7) | 31.0 (7.2) | 32.4 (3.8) | 3.0 (1.6) |
| 15 | 28 (2) | 12 (1) | 0.150 (0.099) | 0.92 (0.10) | 419.1 (105.4) | 39.9 (10.6) | 38.3 (3.5) | 2.6 (0.9) |
| 16 | 24 (1) | 12 (1) | 0.042 (0.009) | 0.76 (0.18) | 256.3 (38.8) | 49.5 (15.8) | 40.2 (2.1) | 5.9 (3.4) |
| 17 | 45 (2) | 9 (2) | - | 0.46 (0.11) | - | 50.9 (7.6) | - | 8.7 (2.8) |
| 18 | 26 (1) | 18 (1) | 0.059 (0.012) | 0.39 (0.20) | 346.8 (45.4) | 123.2 (31.2) | 39.9 (3.3) | 8.7 (2.6) |
| 19 | 36 (2) | 14 (2) | 0.078 (0.022) | 1.40 (0.09) | 204.9 (52.3) | 22.8 (0.4) | 33.5 (3.4) | 1.0 (0.1) |
| 20 | 29 (1) | 9 (1) | 0.118 (0.001) | 0.65 (0.31) | 252.9 (76.6) | 76.1 (28.4) | 29.9 (4.4) | 8.6 (4.9) |

6    **Figure 1**

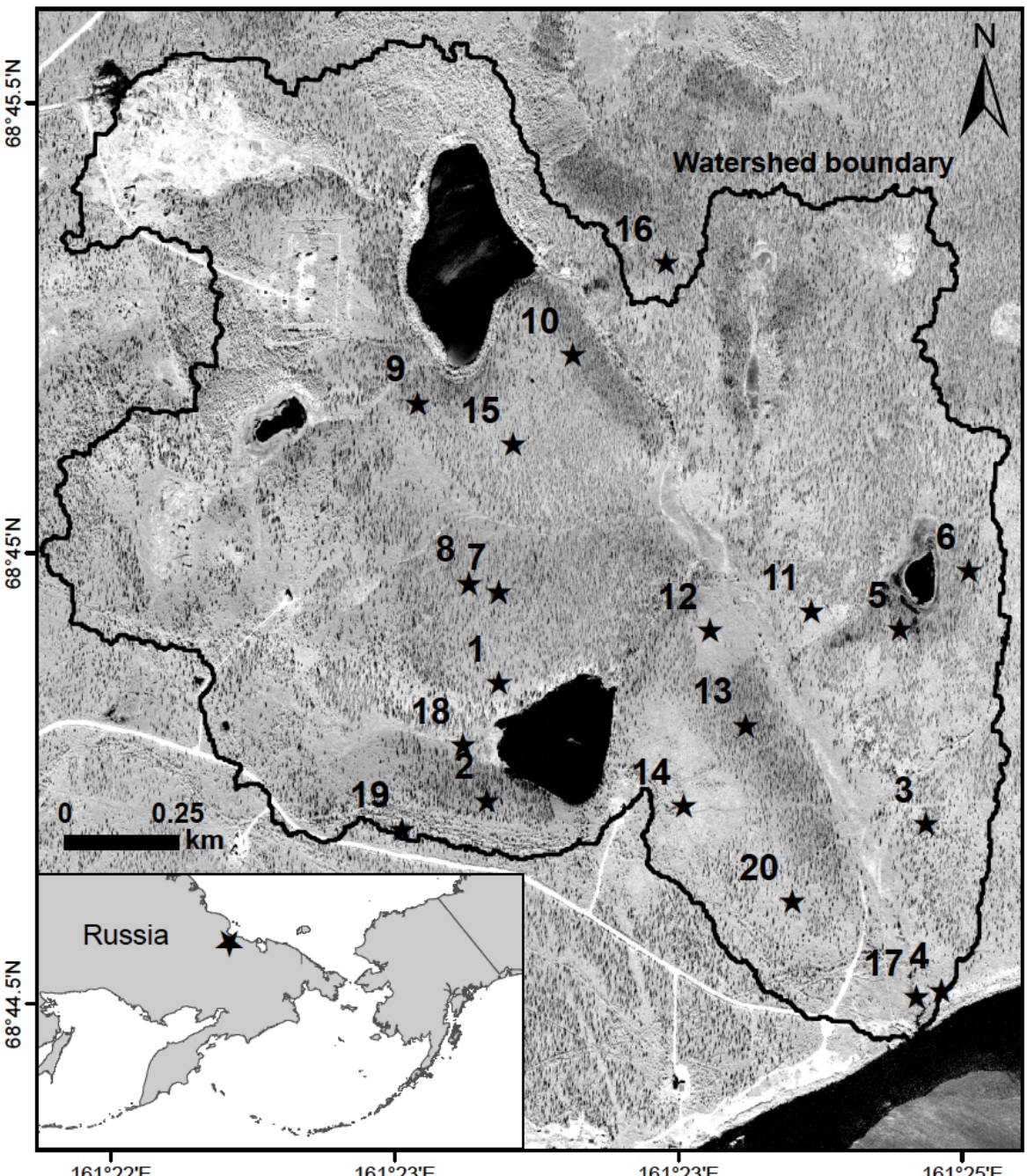

9    **Figure 2**

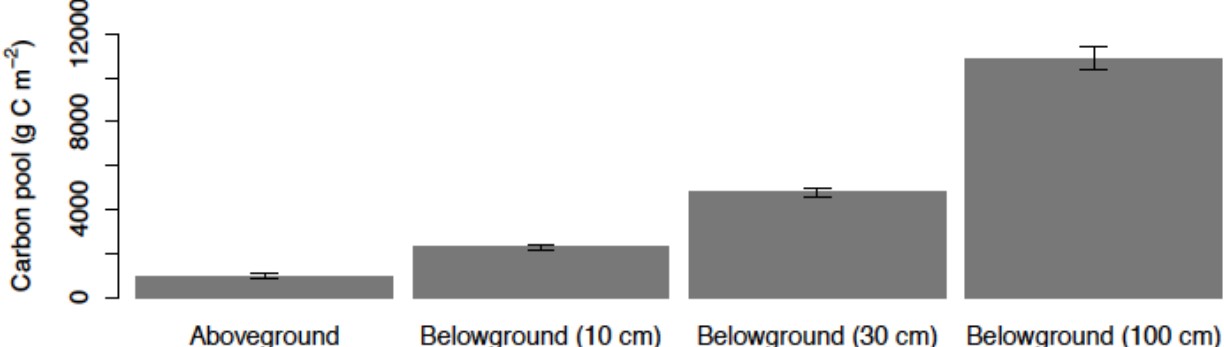

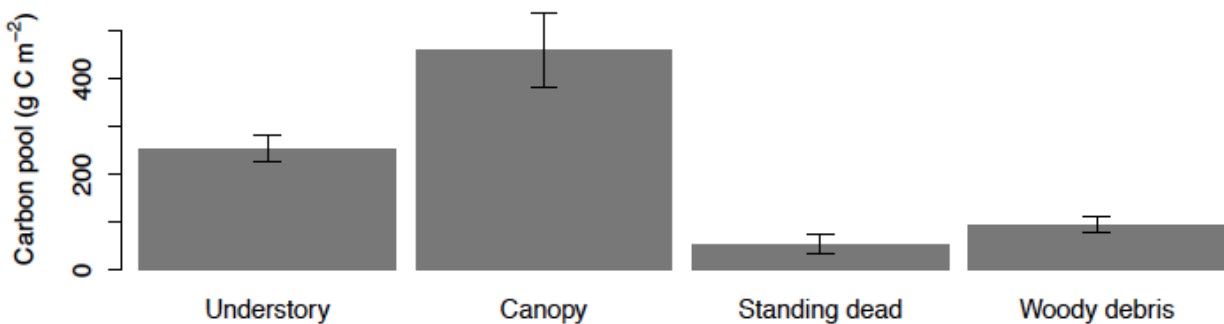


**Figure 3**

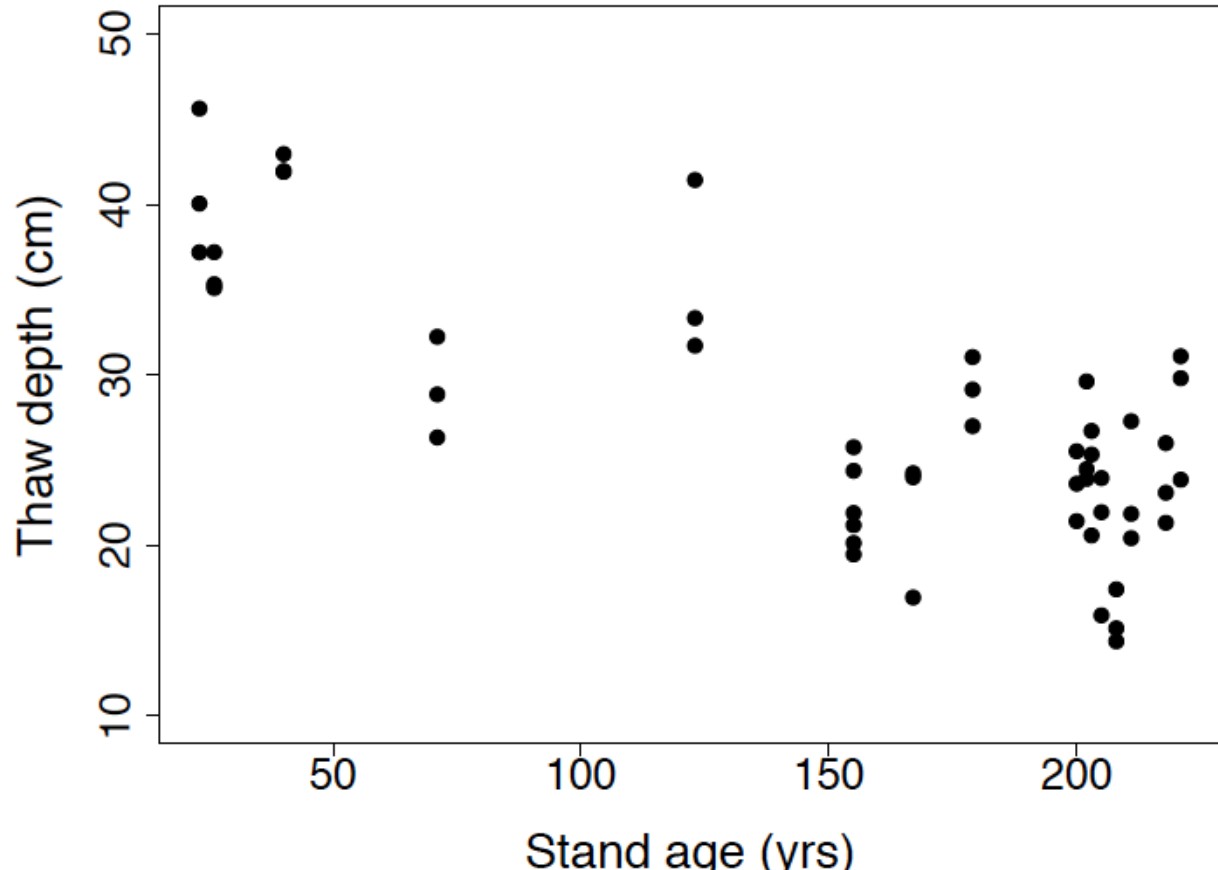

**Figure 4**

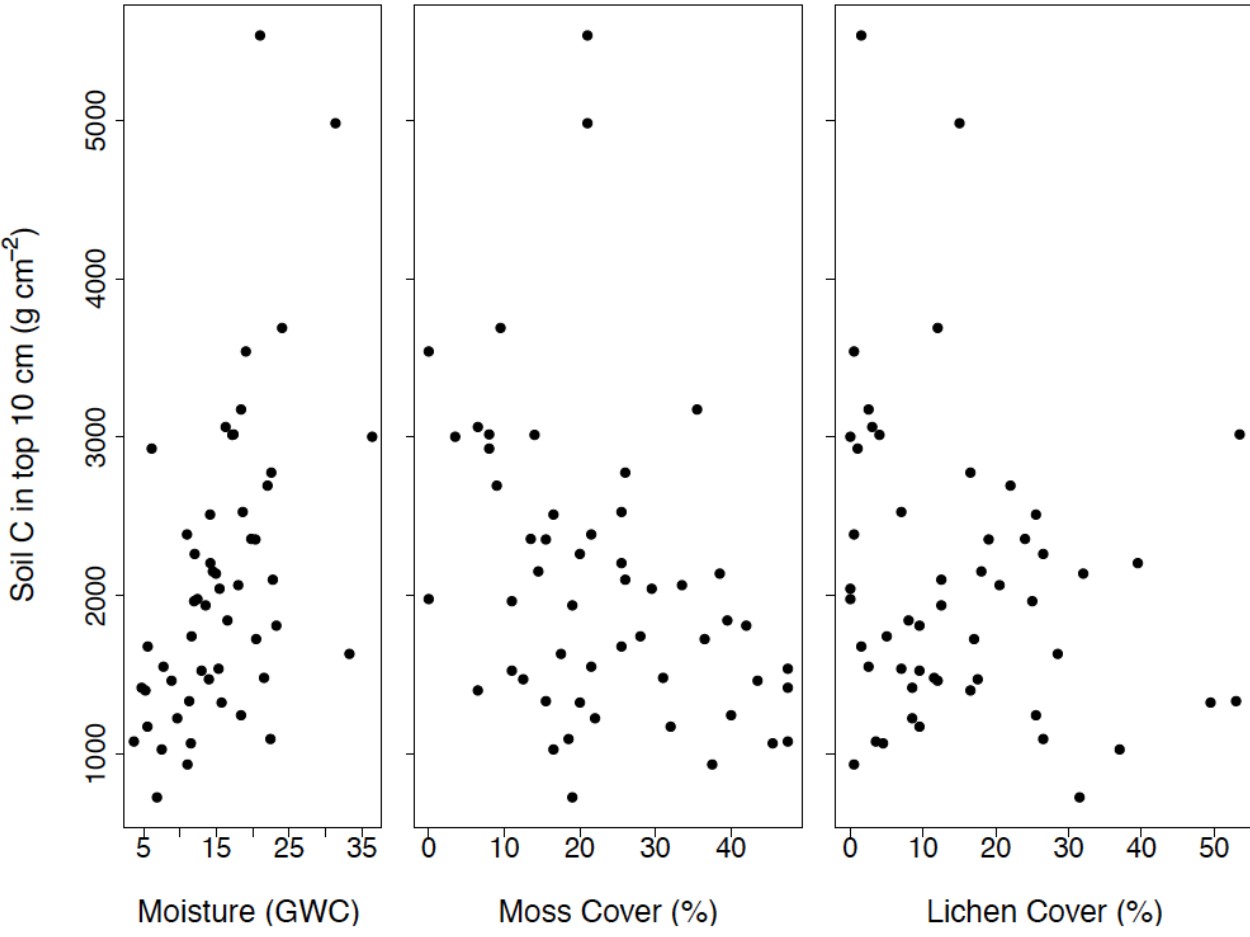

**Figure 5**

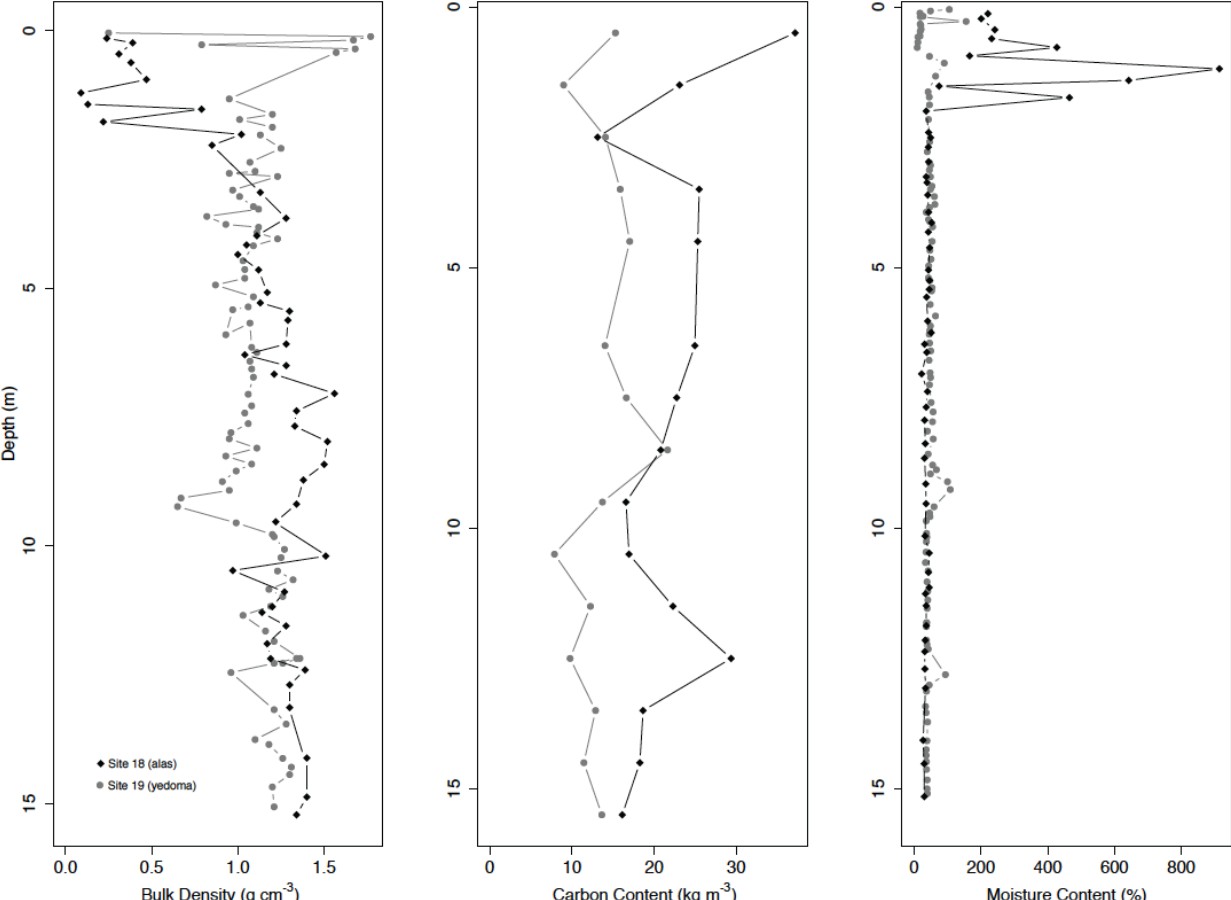

0

