# Peer review of "Elizabeth E. Webb"

_Biogeosciences, 2017_

## Referee Comment (RC1) · Anonymous Referee #1 · 5 May 2017

**Overall Evaluation:**

The manuscript by Webb, Heard and Natali et al. presents an interesting and detailed study on the variability of carbon storage above and belowground in northeast Siberia. The authors address an important and understudied topic showing the tremendous regional carbon variability and this knowledge is needed to increase our understanding of permafrost-climate feedback on global warming. The amount of produced data is substantial, especially for aboveground biomass. The data was thoroughly analyzed with meaningful and valuable results. In general, I like the design and the analyses of this study. The text is well structured and fluent written.

General comments:

It is clear throughout the text that the main emphasis was on the aboveground biomass.

What I lack is the same accuracy and description for the fewer belowground carbon samples, especially for the 7 surface permafrost cores. As a reader I want to know for example: was the coring and the analysis of the 60 cm cores in short increments? How were the 7 sampling sites selected? Any signs of cryoturbation, data on soil texture, etc. Why do you think there is so little carbon in your top meter compared to the results from many other studies? In addition, I suggest that you add the SOC data from the first meter from the two deep cores which are part of the watershed to the other permafrost cores. This additional data will most likely increase the 1m average which will then be similar to many other cited studies. You have nice supplementary data but I lack the information from the permafrost cores in the data.

Since the samples were also analyzed for nitrogen (Line 223), why did you not further incorporate this data in the text? Also, given the sampling and measurement uncertainties, I think is unnecessary to present the soil C values in grams, especially since you shift to kg from line 308 in the text.

**Specific Comments**

- Line 288: Comma used for decimals
- Line 296: I suppose the SE should be  $\pm ?$
- Line 346: Please remove the word "slightly".

- Line 385: How many permafrost soils were sampled: 21 or 7? I miss this information in the section 2.7. Since it is stated in the description of Table 4 "...at selected sites, but not on the transects..."?

- Line 640: Typo "Author(s)"

- Table 2 & 3: Site number 18 is not forested as stated in table 1, why are there values for larch/larch density?

- Table 4: Would be good to indicate the n for the values since they are different
**- Table 4 & 5: Please add in the title "of the mean" as in others tables**

---

## Referee Comment (RC2) · Anonymous Referee #2 · 11 May 2017

This article by Webb and colleagues provides insights in the above and belowground carbon storage at a site in Far East Russia. The site is forested with larch stands and located on Yedoma. The authors present interesting data and find a high variability in C stocks, which they attribute primarily to variations in vegetation.

Overall this is a well written manuscript and there is no doubt that the authors have done a good job at in the writing process. The amount of carbon stored in permafrost regions is an important topic and valuable field data as presented in this article (contrasting above and belowground C stocks) deserves more attention.

The weak points in the manuscript are in my opinion a somewhat confusing sampling scheme or its description, and an underdeveloped discussion that does not challenge the perspective of the authors. In particular, the authors see vegetation as a primary

driver for total C storage, despite the fact that the vast majority of the C is stored in soil and moisture is identified as a major driver of C stocks. To round up the discussion, the authors should also consider that vegetation is merely a reaction to ground conditions and soil forming processes or topographic drivers. Further there is clearly a bias towards the description of vegetation analysis, while the description of the soil sampling and the discussion on soil related aspects is underdeveloped. Please rewrite the sampling description and/or provide a graph outlining the sampling procedure. This is important, because the C variability is one of your major conclusions.

Minor comments:

- L 22 What is snag?
- L 23- 24 rephrase

L 45 - 50 How about thermokarst?

L 58 See also Vitharana et al. (2017) AGU:bgs

L 63 – 65 What do you mean by high resolution sampling and what does this have to do with circumpolar estimates? Also, Walter Anthony et al. (2014) is a paper on thermokarst lake deposits and C accumulation over the Holocene and has nothing to do with soil.

L 70 Yedoma is a sedimentological Suite and not soil, or do you mean the soil developed on top of these Yedoma deposits?

L 72 25m: clearly you cite a number that in Tarnocai et al. 1999 is cited as Zimov et al. 2006  $\rightarrow$  then cite Zimov et al 2006, or find a more up-to-date number

Section 2.2

I am sorry, but this section is a bit confusing.

Add reference to Fig 1.

Did you sample random? If not, then please justify why not and how this could be a bias in your study.

L 142 what is the logic behind this? Please explain.

L 145 – 147 Please rewrite and provide a figure that explains your sampling scheme.

Section 2.3 what is the motivation for this?

L 171 Did you correct your allometric functions for reduced C content in decomposed dead trees? (see for instance Smith et al 2003 GTR report:Forest volume-to-biomass models ...)

L 193 Are these values also valid for Larch trees?

L 218 What soil types did you encounter? How did you select the sampling location with regard to microtopography. Did you have hummocks in the soil? See also Ping (2013) Soil Horiz.

L 213 Please provide more precise constraining dates for the active layer thickness

L 222 again, it is very unclear how you sampled this and how many samples and soil profiles go into one site. This is important to be clarified because an important part of your discussion and your conclusions are based on the variability of these values. What do you mean by 6 samples one at each end of a transect?

L 224 If you only collected the top 10 cm of mineral soil you have a bias towards C enriched upper soil. This can be problematic if you interpolated to deeper depths. If this is the case, please discuss this and outline potential impacts on your statistics.

L 239 Which guidelines did you follow for this?

L 297 and 301 Please use the same units for masses throughout the article. I suggest kg C m-2

L 304 what could this variability be related to?

L 319 Do you mean you started sampling at 0 cm from the top to 10 cm depth or the top 10 cm of the mineral soil?

L 352 I don't see this.

L 406 Also have a look at Siewert et al (2015) AGU:bgs for a comparable study to yours.

L 407 What explains this high variation in your case?

L 420 Please mention that Yakutia spans over a large area with many ecosystem types.

L 428 again, Yedoma is not a soil type

L 448 What do you mean by geophysical controls? You should also discuss these and support your discussions with literature as you did for vegetation related dynamics.

L 459 Please also consider the notion that moist sites support more vegetation that is more productive and stores more C, rather than vegetation driven differences in moisture and thus C

L 471 related -to- stand age

Fig 2: Organic Layer stocks would also be interesting

Table 4 What do you mean by soil classification? Mineral or organic? Or soil type (Podsol, etc...)

Please use the same units thorough the paper! Here it is g Cm -2 before it was kg cm-2

Why is the standard error the same for both columns of the permafrost cores? Are the permafrost cores also including the active layer?

---

## Author Comment (AC1)

We appreciate the input from the editor and from both reviewers, which we feel has made this a stronger manuscript. Our responses to the reviewers' comments follow each comment and are in italicized font. The line numbers referenced in our responses refer to the updated manuscript text, which is attached.

There were some comments by reviewer #2 that seem to have mismatched line numbers (based on the submitted version of the manuscript). We were able to resolve many of these comments, but for some comments it was not clear what text the comments referred. We would be grateful for an opportunity to address these comments after further clarification.
Best,
Sue Natali

**Responses to Reviewer 1**
It is clear throughout the text that the main emphasis was on the aboveground biomass. What I lack is the same accuracy and description for the fewer belowground carbon samples, especially for the 7 surface permafrost cores. As a reader I want to know for example: was the coring and the analysis of the 60 cm cores in short increments?
*The text has been edited at lines 227 to clarify that the cores were sectioned into ~10cm increments, and a Supplement Table has been added with depth increment data.*

How were the 7 sampling sites selected?
*The seven sites where surface permafrost was sampled were a subset of the 20 sampled stands; these seven sites were selected based on accessibility and distribution across the catchment. The 20 stands (i.e. 'sites') were selected to span a range of tree aboveground biomass, as inferred from tree shadows mapped using high-resolution (50 cm) WorldView-1 satellite imagery (Lines 134-136).*

Any signs of cryoturbation, data on soil texture, etc.
*We did not collect data on cryostructure or texture, unfortunately.*

Why do you think there is so little carbon in your top meter compared to the results from many other studies?
*We noted in lines 382-382 that these soil C pool estimates fall within the range of published assessments that characterize this area (i.e., forested area around Cherskiy). However, they are at the low end of the larger region, although within one SD of the regional mean. This may be a result of variation in parent material, disturbance (fire or harvest), or other soil conditions. This assessment, however, is beyond the scope of this study.*

In addition, I suggest that you add the SOC data from the first meter from the two deep cores which are part of the watershed to the other permafrost cores. This additional data will most likely increase the 1m average which will then be similar to many other cited studies.
*We added the 1m SOC data from the two deep cores to the average SOC value presented in the text (line 333-334). In the text, we presented average (+-SE) SOM both with just*

*the yedoma deep core added (because much of the manuscript focuses on yedoma C pools), and then with both the yedoma and alas deep core data added. Figure 2 has been updated with the additional data from the deep yedoma core.*

You have nice supplementary data but I lack the information from the permafrost cores in the data.
*These have been added in Supplement Table 4.*

Since the samples were also analyzed for nitrogen (Line 223), why did you not further incorporate this data in the text?
*We were only able to analyze a subset of soils for C and N because of challenges of transporting international soils. We were able to extrapolate %C, based on C-SOM relationship, to the full dataset, but not possible for %N. Inclusion of N analyses in the methods section was done in error, and we have removed this text.*

Also, given the sampling and measurement uncertainties, I think is unnecessary to present the soil C values in grams, especially since you shift to kg from line 308 in the text.
*Agree. The soil units in the text are in kg, and Table 4 now also is in kg C/m2.*

Specific Comments
- Line 288: Comma used for decimals
*These actually should be commas, not decimals. No change made.*

- Line 296: I suppose the SE should be ±?
*Yes. We have corrected.*

- Line 346: Please remove the word "slightly".
*Done.*

- Line 385: How many permafrost soils were sampled: 21 or 7? I miss this information in the section 2.7. Since it is stated in the description of Table 4 ". . .at selected sites, but not on the transects. . ."?
*We collected three cores at 7 sites for a total of 21 'surface permafrost cores. We edited the table description to read: " Permafrost cores were sampled to 1 m at 7 sites (3/site). ", and clarified the number of samples per site in section 2.8 on line 212.*

- Line 640: Typo "Author(s)"
*Corrected-thanks.*

- Table 2 & 3: Site number 18 is not forested as stated in table 1, why are there values for larch/larch density?
*We corrected table 1 description to read: " All sites were in forested areas except #17 (riparian); Site #18 (alas) had few scattered trees located along one end of the transects."*

- Table 4: Would be good to indicate the n for the values since they are different
*We have indicated the sample size in the table description.*

- Table 4 & 5: Please add in the title "of the mean" as in others tables
*Added.*

**Responses to Reviewer #2**

The weak points in the manuscript are in my opinion a somewhat confusing sampling scheme or its description, and an underdeveloped discussion that does not challenge the perspective of the authors. In particular, the authors see vegetation as a primary driver for total C storage, despite the fact that the vast majority of the C is stored in soil and moisture is identified as a major driver of C stocks. To round up the discussion, the authors should also consider that vegetation is merely a reaction to ground conditions and soil forming processes or topographic drivers. Further there is clearly a bias towards the description of vegetation analysis, while the description of the soil sampling and the discussion on soil related aspects is underdeveloped. Please rewrite the sampling description and/or provide a graph outlining the sampling procedure. This is important, because the C variability is one of your major conclusions.

*The two main suggestions of this reviewer focus on description of the soil sampling and discussion of the drivers of soil C stocks. To address these concerns:*
1. *We edited the methods section and added a supplemental figure, Figure S1.*
2. *Text discussion of soil moisture effects and reference can be found on lines 442-445*

Minor comments:
L 22 What is snag?
*A snag is standing dead or dying tree.*

L 23- 24 rephrase
*Done. Sentence now reads: "Thaw depth was negatively related to stand age, and soil C density (top 10 cm) was positively related to soil moisture and negatively related to moss and lichen cover."*

L 45 – 50 How about thermokarst?
*We changed 'microtopography' to 'topography', and one of the references following that is a thermokarst reference.*

L 58 See also Vitharana et al. (2017) AGU:bgs
*Thanks for suggesting. We added this reference to this manuscript and changed that sentence to read: " Furthermore, permafrost regions are characterized by high heterogeneity in soil C stocks due to variability in soil-forming factors (Vitharana et al., 2017) and at small spatial scales due to cryogenic processes (i.e., cryoturbation at the sub-meter scale)."*

L 63 – 65 What do you mean by high resolution sampling and what does this have to do with circumpolar estimates? Also, Walter Anthony et al. (2014) is a paper on thermokarst lake deposits and C accumulation over the Holocene and has nothing to do with soil.
*By high resolution we mean that spatial resolution of the sampling should match the spatial resolution of the variability. We edited the sentence, and we deleted the Walter Anthony reference.*

L 70 Yedoma is a sedimentological Suite and not soil, or do you mean the soil developed on top of these Yedoma deposits?
*We changed 'soil' to 'deposits'.*

L 72 25m: clearly you cite a number that in Tarnocai et al. 1999 is cited as Zimov et al. 2006 ! then cite Zimov et al 2006, or find a more up-to-date number
*If you are referring to the reference to Tarnocai 2009 on line 66 then we have made the suggested change. If that is not correct, then please clarify and we will make further changes as suggested.*

Section 2.2
I am sorry, but this section is a bit confusing. Add reference to Fig 1.
*Agree. First, we moved the stand age sampling into its own section and moved the stand age and density results into the results section. We cleaned up and clarified the rest of the text in this section. We also added in reference to Figure 1.*

Did you sample random? If not, then please justify why not and how this could be a bias in your study.
*We clarified section 2.2 to note that sites were selected based on biomass distribution; while plots within sites were established based on slope or N-S direction to avoid bias.*

L 142 what is the logic behind this? Please explain.
*I think there is some confusion regarding line numbering; please clarify so that we can address this comment.*
*Lines 141-143 read: " Wood samples were dried at 60 °C and then sanded sequentially with finer grit sizes to obtain a smooth surface. Each sample was then scanned and the annual growth rings were counted using WinDendro (Regent Instruments, Inc., Ontario)."*

L 145 – 147 Please rewrite and provide a figure that explains your sampling scheme.
*We added a supplemental figure and edited the text.*

Section 2.3 what is the motivation for this?
*To explore effects of slope and solar insolation on soil C pools.*

L 171 Did you correct your allometric functions for reduced C content in decomposed dead trees? (see for instance Smith et al 2003 GTR report:Forest volume-to-biomass models ...)
*We did not for snags but did for downed dead trees (line 183-185). Dead standing larch had little observable decay.*

L 193 Are these values also valid for Larch trees?
*We used value for similar structured trees, following methods in previously published studies, as cited; ideally, if available, we would use for larch.*

L 218 What soil How did you select the sampling location
with regard to microtopography. Did you have hummocks in the soil? See also Ping
(2013) Soil Horiz.
There were no hummocks at these locations. Soils were sampled at either end of each of
the three transects (line 213-214; Figure S1) so they were distributed across each site at
~10m distance.

L 213 Please provide more precise constraining dates for the active layer thickness
*We added dates to the text at line 204.*

L 222 again, it is very unclear how you sampled this and how many samples and soil
profiles go into one site. This is important to be clarified because an important part
of your discussion and your conclusions are based on the variability of these values.
What do you mean by 6 samples one at each end of a transect?
*We edited the soil sampling and analysis section to clarify and added a Supplemental
figure.*

L 224 If you only collected the top 10 cm of mineral soil you have a bias towards C
enriched upper soil. This can be problematic if you interpolated to deeper depths. If
this is the case, please discuss this and outline potential impacts on your statistics.
*We did not interpolate. At the 7 sites where we sampled frozen soils, we collected the full
mineral soil profile (lines 217-218) as well as frozen soil. We only used these deeper
samples for the deeper estimates.*

L 239 Which guidelines did you follow for this?
*I think there is some confusion regarding line numbering; please clarify so that we can
address this comment.*
*Lines 238-239 on the submitted manuscript read: "For the deep permafrost samples, sub-
samples used for %C, %OM, and BD measurements were collected from adjacent depth
increments"*

L 297 and 301 Please use the same units for masses throughout the article. I suggest
kg C m-2
*All soil units have been changed to kg C $m^{-2}$.*

L 304 what could this variability be related to?
*Please clarify the line number or specify the text that you are referencing.*

L 319 Do you mean you started sampling at 0 cm from the top to 10 cm depth or the
top 10 cm of the mineral soil?
*We are referring to the top 10 cm of the ground surface, not the top of the mineral soil.
We have clarified this in the text (lines 322-323) to read: "Soil C density in the top 10 cm
of the ground surface (0-10 cm soil depth, which may have contained both organic and
mineral soils)…"*

L 352 I don't see this.

*Please clarify the line number or text this is referencing; it's not clear what changes are suggested to line 352.*
*Line 350-352 read: "In addition, our larch AGB estimates fell within the low range of larch stands across other high-latitude (> 64° N) regions and were generally 3-10 times lower than other stands (Kajimoto et al., 2010) "*

L 406 Also have a look at Siewert et al (2015) AGU:bgs for a comparable study to yours.
*Thank you for the suggestion. Reference has been incorporated at line 425.*

L 407 What explains this high variation in your case?
*Assuming this refers to line 394, I don't think we have enough samples/information to conduct this analysis, but much of the variation may have been driven by high and variable ice content.*

L 420 Please mention that Yakutia spans over a large area with many ecosystem types.
*We added text to note that the region comprises a diverse range of ecosystem types.*

L 428 again, Yedoma is not a soil type
*We have corrected throughout the text.*

L 448 What do you mean by geophysical controls?
*We changed 'geophysical' to 'parent material and climate'. These factors were not the focus of the discussion as the sites were located within a small catchment with similar parent material and climate.*

L 459 Please also consider the notion that moist sites support more vegetation that is more productive and stores more C, rather than vegetation driven differences in moisture and thus C
*We edited the text at lines 442-445 to address this comment and added a reference to Berner et al (2013).*

L 471 related -to- stand age
*Corrected.*

Fig 2: Organic Layer stocks would also be interesting
*Organic layer carbon stocks are provided in Figure 4.*

Table 4 What do you mean by soil classification? Mineral or organic? Or soil type (Podsol, etc...)
*We clarified to read "soil type (mineral/organic)"*

Please use the same units thorough the paper! Here it is g Cm -2 before it was kg cm-2
*All soils are now in units of kg C m-2.*

Why is the standard error the same for both columns of the permafrost cores? Are the permafrost cores also including the active layer?

*The SEs were an error, which have now been corrected--thank you for catching this. The columns under 'thawed soil cores' are thawed active layer. The permafrost core data presented are C pools in the top 0-30cm of ground or C pools in the top 100 cm of ground. We edited the table description to clarify.*

[revised manuscript text omitted]

C can accumulate at a higher rate than in more well-drained, well-aerated soils (Schädel et al.,

2016).  However, it is likely that this positive association between moisture and C density may also be a result of increased C inputs and plant productivity associated with higher soil moisture (Berner et al. 2013) or the lateral movement of dissolved organic C into the wetter sites.

 Species composition also plays an important role in soil C storage in boreal forests (Hollingsworth et al., 2008) through the quality and quantity of litter inputs and their effects on environmental controls such as soil moisture and temperature.  Lichens and mosses are sometimes thought to encourage soil C storage through their promotion of low soil temperatures, higher moisture, and a relatively acidic environment (Bonan and Shugar, 1989).  However, at our sites, increasing abundance of lichen and moss was associated with lower soil C storage, which may have been due to lower rates of C fixation (Turetsky et al., 2010), higher rates of decomposition of vascular plant litter in moss and lichen patches (Wardle et al., 2003), or impacts of moss and lichen on soil moisture and soil temperatures.

 Increasing thaw depth may result in increased C loss from boreal ecosystems; as more soil is thawed, more organic matter is available for decomposition.  We found that thaw depth was negatively related to stand age, which is likely because forest fires tend to increase thaw depth (O'Donnell et al., 2011; Yoshikawa et al., 2002) and the deeper thaw depth observed in the younger sites could be a result of more recent burning events.

**5  CONCLUSIONS**

We found that the overwhelming majority of C in the Y4 watershed was stored belowground but that the amount of C within any given pool was highly variable throughout the landscape; C storage in AGB varied up to 95% among sites and there was 69% variation in the top 10 cm of soil, 36% in the top 30 cm, and 28% in the top 1 m.   This variability among sites in our study was similar to the variability between our sites and others that were 3 to 50 km away (Matsuura et al., 2005; Palmtag et al., 2015), indicating a high level of natural variability at the meter and kilometer scales.  Our results also indicate higher soil C variability in surface soils when compared to deeper soils, indicating that recent, on-going processes significantly contribute to soil C variability.  Specifically, our results suggest that aboveground processes such as the regulation of soil moisture by aboveground vegetation, vegetation community structure and litter inputs are influential in controlling 
[revised manuscript text omitted]

| 1 | 23 (1) | 13 (1) | 0.078 (0.021) | 0.52 (0.16) | 198.9 (34.4) | 64.7 (17.4) | 37.6 (3.5) | 6.9 (2.5) |
| 2 | 22 (1) | 11 (1) | 0.040 (0.011) | 0.64 (0.05) | 203.8 (28.0) | 33.9 (5.8) | 38.3 (4.1) | 2.4 (0.5) |
| 3 | 24 (1) | 14 (1) | 0.062 (0.011) | 0.70 (0.11) | 103.3 (16.2) | 29.1 (4.4) | 30.4 (2.2) | 2.3 (0.6) |
| 4 | 41 (2) | 10 (1) | 0.148 (0.063) | 0.54 (0.14) | 107.3 (28.9) | 61.0 (15.6) | 26.6 (4.0) | 8.7 (3.0) |
| 5 | 23 (1) | 8 (1) | 0.120 (0.032) | 1.02 (0.08) | 220.2 (23.1) | 25.6 (2.1) | 39.2 (3.2) | 1.6 (0.3) |
| 6 | 21 (2) | 9 (1) | 0.113 (0.039) | 0.63 (0.05) | 182.0 (19.8) | 34.2 (6.1) | 39.0 (3.0) | 3.8 (1.0) |
| 7 | 21 (1) | 12 (1) | 0.026 (0.005) | 0.76 (0.18) | 348.5 (48.4) | 43.6 (10.2) | 44.4 (2.0) | 3.9 (1.2) |
| 8 | 16 (1) | 11 (1) | 0.027 (0.002) | 0.68 (0.10) | 304.9 (32.1) | 46.4 (10.3) | 46.7 (0.6) | 4.4 (1.1) |
| 9 | 26 (2) | 13 (1) | 0.082 (0.010) | 0.64 (0.12) | 171.3 (29.5) | 46.5 (11.2) | 30.9 (4.4) | 5.5 (2.1) |
| 10 | 23 (1) | 11 (1) | 0.048 (0.007) | 0.89 (0.05) | 272.6 (15.2) | 26.5 (1.7) | 43.6 (1.9) | 1.6 (0.2) |
| 11 | 35 (2) | 10 (1) | 0.060 (0.023) | 0.84 (0.12) | 142.8 (17.8) | 39.4 (6.9) | 30.5 (3.3) | 3.6 (1.6) |
| 12 | 29 (2) | 10 (1) | 0.053 (0.020) | 0.67 (0.10) | 247.7 (17.5) | 58.3 (10.7) | 43.5 (1.8) | 5.0 (1.0) |
| 13 | 29 (1) | 12 (1) | 0.042 (0.008) | 0.71 (0.11) | 194.1 (15.4) | 48.6 (12.6) | 40.0 (1.4) | 4.0 (1.0) |
| 14 | 42 (2) | 8 (1) | 0.103 (0.016) | 0.82 (0.10) | 165.8 (14.7) | 31.0 (7.2) | 32.4 (3.8) | 3.0 (1.6) |
| 15 | 28 (2) | 12 (1) | 0.150 (0.099) | 0.92 (0.10) | 419.1 (105.4) | 39.9 (10.6) | 38.3 (3.5) | 2.6 (0.9) |
| 16 | 24 (1) | 12 (1) | 0.042 (0.009) | 0.76 (0.18) | 256.3 (38.8) | 49.5 (15.8) | 40.2 (2.1) | 5.9 (3.4) |
| 17 | 45 (2) | 9 (2) | - | 0.46 (0.11) | - | 50.9 (7.6) | - | 8.7 (2.8) |
| 18 | 26 (1) | 18 (1) | 0.059 (0.012) | 0.39 (0.20) | 346.8 (45.4) | 123.2 (31.2) | 39.9 (3.3) | 8.7 (2.6) |
| 19 | 36 (2) | 14 (2) | 0.078 (0.022) | 1.40 (0.09) | 204.9 (52.3) | 22.8 (0.4) | 33.5 (3.4) | 1.0 (0.1) |
| 20 | 29 (1) | 9 (1) | 0.118 (0.001) | 0.65 (0.31) | 252.9 (76.6) | 76.1 (28.4) | 29.9 (4.4) | 8.6 (4.9) |

**Table S4.** Soil characteristics of surface permafrost cores (frozen active layer and surface permafrost; type=F) and thawed active layer mineral soils (type=TM) in the Y4 watershed. Depths reflect distance from the top of the mineral layer. Soil carbon pools are reported for each depth increment. Active layer organic soil data are in Table S2.

| Site | Core | Type | Depth (cm) | Organic Matter Content (%) | Carbon Content (%) | Gravimetric Water Content (%) | Bulk Density (g cm$^{-3}$) | Soil C (kg m$^{-2}$) |
|------|------|------|------------|----------|----------|----------|----------|----------|
| 1 | 1 | TM | 0-28 | 3.66 | 1.34 | 24 | 1.04 | 3.92 |
| 1 | 1 | TM | 28-51 | 2.77 | 0.88 | 24 | 0.84 | 1.69 |
| 1 | 1 | F | 51-60 | 2.58 | 0.78 | 78 | 0.93 | 0.72 |
| 1 | 1 | F | 60-70 | 2.60 | 0.79 | 96 | 0.88 | 0.70 |
| 1 | 1 | F | 70-82 | 3.04 | 1.02 | 198 | 0.42 | 0.51 |
| 1 | 1 | F | 82-92 | 2.87 | 0.93 | 116 | 0.60 | 0.56 |
| 1 | 1 | F | 92-102.5 | 2.67 | 0.82 | 109 | 0.55 | 0.48 |
| 1 | 1 | F | 102.5-108.5 | 2.61 | 0.79 | 117 | 0.63 | 0.79 |
| 1 | 2 | TM | 0-26 | 3.80 | 1.42 | 25 | 0.83 | 3.05 |
| 1 | 2 | TM | 26-39 | 3.15 | 1.07 | 24 | 1.12 | 1.57 |
| 1 | 2 | F | 39-50 | 2.51 | 0.74 | 69 | 0.84 | 0.68 |
| 1 | 2 | F | 50-60 | 2.50 | 0.74 | 127 | 0.67 | 0.50 |
| 1 | 2 | F | 60-71 | 2.24 | 0.60 | 72 | 0.93 | 0.61 |
| 1 | 2 | F | 71-81 | 2.39 | 0.68 | 89 | 0.78 | 0.53 |
| 1 | 2 | F | 81-91 | 2.31 | 0.63 | 76 | 0.81 | 0.52 |
| 1 | 2 | F | 91-102 | 2.67 | 0.83 | 68 | 0.91 | 0.83 |
| 1 | 2 | F | 102-112 | 2.69 | 0.84 | 77 | 0.85 | 0.71 |
| 1 | 2 | F | 112-121 | 3.04 | 1.02 | 70 | 0.83 | 0.76 |
| 1 | 3 | TM | 0-27 | 4.15 | 1.60 | 29 | 0.70 | 3.02 |
| 1 | 3 | TM | 27-45 | 2.75 | 0.86 | 22 | 1.08 | 1.67 |
| 1 | 3 | F | 45-50 | - | 1.32 | 32 | 1.13 | 0.66 |
| 1 | 3 | F | 50-60 | 2.11 | 0.53 | 72 | 1.00 | 0.47 |
| 1 | 3 | F | 60-71 | 2.66 | 0.82 | 114 | 0.76 | 0.52 |
| 1 | 3 | F | 71-81 | 2.63 | 0.80 | 79 | 0.87 | 0.67 |
| 1 | 3 | F | 81-90 | 2.73 | 0.85 | 116 | 0.73 | 0.42 |
| 1 | 3 | F | 90-101 | 2.73 | 0.85 | 114 | 0.70 | 0.57 |
| 1 | 3 | F | 101-112 | 2.70 | 0.84 | 96 | 0.61 | 0.59 |
| 2 | 1 | TM | 0-38 | 4.03 | 1.54 | 30 | 0.64 | 3.74 |
| 2 | 1 | F | 38-51 | 5.20 | 2.15 | 50 | 1.20 | 3.35 |
| 2 | 1 | F | 51-60 | 3.69 | 1.36 | 134 | 0.53 | 0.65 |
| 2 | 1 | F | 60-69 | - | 1.26 | 129 | 0.50 | 0.57 |
| 2 | 1 | F | 69-81 | 3.32 | 1.17 | 130 | 0.59 | 0.83 |
| 2 | 1 | F | 81-90 | 2.53 | 0.75 | 108 | 0.73 | 0.49 |
| 2 | 1 | F | 90-99 | 2.52 | 0.75 | 127 | 0.59 | 0.40 |
| 2 | 1 | F | 99-109.5 | 2.24 | 0.60 | 148 | 0.56 | 0.35 |
| 2 | 2 | TM | 0-26 | 2.98 | 0.98 | 22 | 1.00 | 2.57 |
| 2 | 2 | TM | 26-42 | 3.11 | 1.05 | 28 | 0.87 | 1.47 |
| 2 | 2 | F | 42-50 | 2.89 | 0.94 | 66 | 0.97 | 0.73 |
| 2 | 2 | F | 50-60 | 2.76 | 0.87 | 75 | 0.78 | 0.68 |
| 2 | 2 | F | 60-74 | 3.26 | 1.13 | 105 | 0.69 | 1.03 |
| 2 | 2 | F | 74-84 | 2.66 | 0.82 | 89 | 0.85 | 0.69 |

**Table S4.** Soil characteristics of surface permafrost cores (frozen active layer and surface permafrost; type=F) and thawed active layer mineral soils (type=TM) in the Y4 watershed. Depths reflect distance from the top of the mineral layer. Soil carbon pools are reported for each depth increment. Active layer organic soil data are in Table S2.

| | | | | | | | | |
|---|---|---|---|---|---|---|---|---|
| 2 | 2 | F | 84-95 | 2.64 | 0.81 | 105 | 0.70 | 0.62 |
| 2 | 2 | F | 95-104 | 3.78 | 1.41 | 81 | 0.81 | 1.03 |
| 2 | 2 | F | 104-111 | 4.02 | 1.53 | 71 | 0.87 | 0.94 |
| 2 | 3 | TM | 0-29 | 4.60 | 1.83 | 35 | 0.81 | 4.30 |
| 2 | 3 | F | 29-40 | 3.34 | 1.18 | 80 | 1.15 | 1.49 |
| 2 | 3 | F | 40-49 | 2.82 | 0.90 | 169 | 0.46 | 0.37 |
| 2 | 3 | F | 49-60 | 3.31 | 1.16 | 65 | 1.07 | 1.36 |
| 2 | 3 | F | 60-68 | 2.99 | 0.99 | 103 | 0.72 | 0.57 |
| 2 | 3 | F | 68-80 | 3.15 | 1.07 | 101 | 0.90 | 1.16 |
| 2 | 3 | F | 80-90 | 3.31 | 1.16 | 114 | 0.60 | 0.69 |
| 2 | 3 | F | 90-95 | 3.05 | 1.02 | 120 | 0.75 | 0.38 |
| 7 | 1 | TM | 0-23 | 4.09 | 1.57 | 31 | 0.92 | 3.32 |
| 7 | 1 | F | 23-31 | 3.54 | 1.28 | 143 | 0.54 | 0.49 |
| 7 | 1 | F | 31-38 | 3.35 | 1.18 | 372 | 0.31 | 0.17 |
| 7 | 1 | F | 38-51 | 3.34 | 1.18 | 134 | 0.60 | 0.77 |
| 7 | 1 | F | 51-62 | 3.07 | 1.03 | 45 | 1.19 | 1.07 |
| 7 | 1 | F | 62-72 | 2.73 | 0.85 | 69 | 1.01 | 0.76 |
| 7 | 1 | F | 72-81 | 4.20 | 1.63 | 85 | 0.87 | 1.06 |
| 7 | 2 | TM | 0-2.5 | 14.04 | 6.78 | 77 | 0.49 | 0.82 |
| 7 | 2 | F | 2.5-12 | 5.13 | 2.11 | 44 | 1.04 | 2.05 |
| 7 | 2 | F | 12-23 | 4.03 | 1.54 | 109 | 0.76 | 1.00 |
| 7 | 2 | F | 23-33 | 3.46 | 1.24 | 147 | 0.65 | 0.61 |
| 7 | 2 | F | 33-44 | 3.49 | 1.25 | 126 | 0.70 | 0.87 |
| 7 | 2 | F | 44-55 | 3.96 | 1.50 | 62 | 0.81 | 1.35 |
| 7 | 2 | F | 55-67 | 4.44 | 1.75 | 70 | 0.99 | 1.84 |
| 7 | 2 | F | 67-79 | 5.34 | 2.22 | 190 | 0.38 | 0.87 |
| 7 | 2 | F | 79-90 | 4.25 | 1.65 | 225 | 0.34 | 0.54 |
| 7 | 2 | F | 90-103 | 3.01 | 1.00 | 142 | 0.60 | 0.68 |
| 7 | 3 | TM | 0-25.5 | 5.14 | 2.12 | 31 | 0.80 | 4.32 |
| 7 | 3 | TM | 25.5-29 | 3.68 | 1.36 | 24 | 2.71 | 1.28 |
| 7 | 3 | F | 29-40 | 3.96 | 1.50 | 63 | 1.06 | 1.66 |
| 7 | 3 | F | 40-51 | 3.20 | 1.10 | 87 | 0.77 | 0.78 |
| 7 | 3 | F | 51-62 | 3.54 | 1.28 | 267 | 0.31 | 0.39 |
| 7 | 3 | F | 62-71 | 2.40 | 0.68 | 102 | 0.83 | 0.37 |
| 7 | 3 | F | 71-83 | 2.63 | 0.80 | 115 | 0.80 | 0.63 |
| 9 | 1 | TM | 0-19 | 3.65 | 1.33 | 25 | 0.74 | 1.88 |
| 9 | 1 | F | 19-30 | 2.48 | 0.73 | 130 | 0.74 | 0.59 |
| 9 | 1 | F | 30-41 | 2.51 | 0.74 | 53 | 0.82 | 0.67 |
| 9 | 1 | F | 41-53 | 2.66 | 0.82 | 38 | 1.61 | 1.58 |
| 9 | 1 | F | 53-63 | 2.38 | 0.67 | 81 | 0.74 | 0.50 |
| 9 | 1 | F | 63-69 | 1.53 | 0.23 | 70 | 0.95 | 0.13 |
| 9 | 2 | TM | 0-30 | 2.92 | 0.95 | 20 | 1.14 | 3.25 |
| 9 | 2 | TM | 30-41.5 | 2.76 | 0.87 | 21 | 0.44 | 1.59 |
| 9 | 2 | F | 41.5-49 | 1.97 | 0.46 | 65 | 0.94 | 0.32 |

**Table S4.** Soil characteristics of surface permafrost cores (frozen active layer and surface permafrost; type=F) and thawed active layer mineral soils (type=TM) in the Y4 watershed. Depths reflect distance from the top of the mineral layer. Soil carbon pools are reported for each depth increment. Active layer organic soil data are in Table S2.

| | | | | | | | | |
|---|---|---|---|---|---|---|---|---|
| 9 | 2 | F | 49-58 | 2.12 | 0.53 | 83 | 0.92 | 0.44 |
| 9 | 2 | F | 58-67 | 2.14 | 0.54 | 120 | 0.58 | 0.29 |
| 9 | 2 | F | 67-77 | 2.08 | 0.52 | 80 | 0.83 | 0.43 |
| 9 | 2 | F | 77-86 | 2.00 | 0.47 | 100 | 0.69 | 0.29 |
| 9 | 2 | F | 86-95 | 2.16 | 0.56 | 125 | 0.57 | 0.29 |
| 9 | 2 | F | 95-102 | 2.20 | 0.58 | 77 | 0.76 | 0.31 |
| 9 | 3 | TM | 0-33.5 | 2.76 | 0.87 | 23 | 0.77 | 2.25 |
| 9 | 3 | F | 34-40 | 2.41 | 0.69 | 43 | 0.87 | 0.36 |
| 9 | 3 | F | 40-48 | 2.09 | 0.52 | 92 | 0.89 | 0.37 |
| 9 | 3 | F | 48-58 | 2.00 | 0.47 | 197 | 0.78 | 0.37 |
| 9 | 3 | F | 58-68 | 2.10 | 0.52 | 64 | 0.90 | 0.47 |
| 9 | 3 | F | 68-82 | 2.05 | 0.50 | 98 | 0.74 | 0.52 |
| 10 | 1 | TM | 0-3.5 | 4.77 | 1.92 | 43 | 0.76 | 0.51 |
| 10 | 1 | F | 3.5-11 | 2.07 | 0.51 | 42 | 1.18 | 0.44 |
| 10 | 1 | F | 11-20 | 2.67 | 0.82 | 125 | 0.74 | 0.34 |
| 10 | 1 | F | 20-32 | - | 0.85 | 104 | 0.59 | 0.55 |
| 10 | 1 | F | 32-40 | 2.77 | 0.88 | 103 | 0.66 | 0.40 |
| 10 | 1 | F | 40-51 | 2.64 | 0.81 | 96 | 0.64 | 0.67 |
| 10 | 1 | F | 51-65 | 3.86 | 1.45 | 223 | 0.39 | 0.79 |
| 10 | 2 | TM | 0-23.5 | 4.59 | 1.83 | 35 | 0.74 | 3.20 |
| 10 | 2 | F | 23.5-30 | 4.54 | 1.80 | 50 | 1.18 | 1.38 |
| 10 | 2 | F | 30-41 | 2.63 | 0.80 | 91 | 0.76 | 0.67 |
| 10 | 2 | F | 41-51 | 2.31 | 0.64 | 173 | 0.49 | 0.31 |
| 10 | 2 | F | 51-62 | 2.51 | 0.74 | 90 | 0.66 | 0.54 |
| 10 | 2 | F | 62-72 | 2.40 | 0.68 | 212 | 0.29 | 0.20 |
| 10 | 2 | F | 72-81 | 2.20 | 0.58 | 133 | 0.19 | 0.10 |
| 10 | 2 | F | 81-91 | 2.52 | 0.74 | 82 | 0.84 | 0.62 |
| 10 | 3 | TM | 0-2.5 | 7.38 | 3.29 | 47 | 0.66 | 0.54 |
| 10 | 3 | F | 2.5-9 | 2.05 | 0.50 | 57 | 1.01 | 0.32 |
| 10 | 3 | F | 9-19 | 3.80 | 1.42 | 71 | 0.78 | 1.22 |
| 10 | 3 | F | 19-29 | 3.71 | 1.37 | 111 | 0.58 | 0.78 |
| 10 | 3 | F | 29-41 | 4.54 | 1.80 | 233 | 0.31 | 0.73 |
| 10 | 3 | F | 41-51 | 4.93 | 2.01 | 155 | 0.45 | 0.87 |
| 10 | 3 | F | 51-59 | 2.90 | 0.95 | 99 | 0.74 | 0.45 |
| 16 | 1 | TM | 0-33.5 | 2.98 | 0.98 | 23 | 0.54 | 1.79 |
| 16 | 1 | F | 33.5-40 | 3.54 | 1.28 | 48 | 1.00 | 0.86 |
| 16 | 1 | F | 40-50 | 3.16 | 1.08 | 85 | 0.82 | 0.71 |
| 16 | 1 | F | 50-61 | 2.81 | 0.90 | 48 | 1.08 | 1.02 |
| 16 | 1 | F | 61-72 | 3.59 | 1.31 | 84 | 0.53 | 0.87 |
| 16 | 1 | F | 72-81 | 3.12 | 1.06 | 451 | 0.21 | 0.17 |
| 16 | 1 | F | 81-88 | 3.28 | 1.15 | 655 | 0.16 | 0.10 |
| 16 | 2 | TM | 0-23 | 5.35 | 2.23 | 33 | 0.61 | 3.13 |
| 16 | 2 | F | 23-30 | 3.10 | 1.05 | 46 | 1.35 | 0.99 |
| 16 | 2 | F | 30-41 | 3.01 | 1.00 | 253 | 0.29 | 0.32 |

**Table S4.** Soil characteristics of surface permafrost cores (frozen active layer and surface permafrost; type=F) and thawed active layer mineral soils (type=TM) in the Y4 watershed. Depths reflect distance from the top of the mineral layer. Soil carbon pools are reported for each depth increment.  Active layer organic soil data are in Table S2.

| | | | | | | | | |
|---|---|---|---|---|---|---|---|---|
| 16 | 2 | F | 41-52 | 4.07 | 1.56 | 598 | 0.17 | 0.29 |
| 16 | 2 | F | 52-65 | 2.66 | 0.82 | 391 | 0.21 | 0.22 |
| 16 | 2 | F | 65-70 | 3.34 | 1.18 | | 0.07 | 0.04 |
| 16 | 2 | F | 70-80 | 2.86 | 0.92 | 328 | 0.18 | 0.16 |
| 16 | 3 | TM | 0-29 | 14.86 | 4.76 | 22 | 0.86 | 12.54 |
| 16 | 3 | TM | 29-37.5 | 2.79 | 0.89 | 23 | 2.36 | 1.78 |
| 16 | 3 | F | 37.5-49 | 3.43 | 1.22 | 136 | 0.53 | 0.75 |
| 16 | 3 | F | 49-52 | 3.23 | 1.12 | 265 | 0.31 | 0.10 |
| 16 | 3 | F | 52-63 | 5.78 | 2.46 | 168 | 0.54 | 1.09 |
| 16 | 3 | F | 63-75 | 5.82 | 2.48 | 137 | 0.55 | 1.64 |
| 16 | 3 | F | 75-87 | 3.76 | 1.39 | 178 | 0.46 | 0.69 |
| 20 | 1 | TM | 0-12 | 3.30 | 1.15 | 23 | 1.31 | 1.81 |
| 20 | 1 | F | 12-28 | 2.52 | 0.74 | 27 | 0.97 | 1.16 |
| 20 | 1 | F | 28-36 | 2.27 | 0.61 | 22 | 1.92 | 0.95 |
| 20 | 1 | F | 36-46 | 2.26 | 0.61 | 43 | 1.40 | 0.86 |
| 20 | 1 | F | 46-57 | 2.14 | 0.55 | 72 | 0.88 | 0.53 |
| 20 | 1 | F | 57-68 | 2.03 | 0.49 | 149 | 0.54 | 0.29 |
| 20 | 2 | TM | 0-14 | 3.08 | 1.04 | 20 | 2.48 | 3.60 |
| 20 | 2 | TM | 14-24 | 2.35 | 0.66 | 21 | 1.22 | 0.80 |
| 20 | 2 | F | 44-50 | 2.81 | 0.90 | 28 | 1.88 | 1.01 |
| 20 | 2 | F | 50-60 | 2.19 | 0.57 | 62 | 1.04 | 0.60 |
| 20 | 2 | F | 60-70 | 1.93 | 0.43 | 87 | 0.81 | 0.35 |
| 20 | 2 | F | 70-78 | 2.06 | 0.51 | 92 | 0.89 | 0.36 |
| 20 | 3 | TM | 0-22 | 3.17 | 1.09 | 23 | 1.61 | 3.84 |
| 20 | 3 | TM | 22-33 | 3.17 | 1.08 | 23 | 1.45 | 1.73 |
| 20 | 3 | F | 33-40 | 2.06 | 0.51 | 30 | 1.14 | 0.40 |
| 20 | 3 | F | 40-48 | 2.17 | 0.56 | 57 | 1.04 | 0.47 |
| 20 | 3 | F | 48-55 | 2.00 | 0.47 | 25 | 1.63 | 0.54 |
| 20 | 3 | F | 55-65 | 2.42 | 0.69 | 124 | 0.71 | 0.49 |

---

## Author Response (AR1)

9 August 2017

Dear Dr. Subke,

Thank you for your helpful comments and edits. Please find our responses below in italics following your comments and on the attached revised manuscript, which has this last round of edits tracked. Please let me know if additional changes are needed. Thank you for your time.

Sincerely,
Sue Natali

**Responses to editor's comments:**
I have a remaining issue that has to be clarified before this is publishable (also raised by referee 2). There is some circularity in arguing that vegetation and/or moisture drive belowground C stores, as these parameters are all linked. Correlations are not clearly supported statistically (please see my comments on lines 327/328 below), and you should discuss more carefully the possibility that explaining soil C stores by vegetation and moisture alone is likely to be simplistic.
*We clarified the discussion regarding the patterns of soil C storage and its relationship to soil moisture and vegetation, and adjusted text throughout to clarify that these environmental variables are interconnected and that soil C is not likely a simple function of a single environmental control or a single directional effect.*

264-266: I can't see how you can apply a parametric test with a sample size of 1. Using depth increments within 1 m sections is at best pseudo replication, and not a robust method to determine soil parameters, when true replication is n=1 per soil type. Your figure 5 shows no error bars (which is I think correct), but you can't present differences between these two profiles as either statistically significantly different or not.
*We agree. We have removed the statistical comparison from the manuscript. Text was deleted from the methods and from the results.*

305: Remove space after commas within figures for tree biomass.
*Done.*

307: Do you mean "mostly highly correlated", or "showed the highest (or best?) correlation"?
*Corrected to read: "showed the highest correlation".*

327/328: There is a tendency to treat significant correlation between scalars as explanatory variables (distinction of correlation an causation). Here, you suggest that moss and lichen cover "control" soil C content in the top 10 cm, even though the causal link is not clear. It's possible that soil moisture influences both

parameters creating the observed correlation without a direct dependence of these on one another. I also miss an indication of whether the correlations with moisture and moss/lichen cover were significant. Table S3 gives parameter values, but no statistical significance. From Figure 4, it looks as though any correlation would be weak. This is a central point in your results, and has to be supported robustly by statistics. If the correlation is weak (or indeed not significant), then you can not make some of the statements (including in the abstract).

*We changed these lines to read: "The distribution of soil C density in the top 10 cm was best explained by soil moisture, percent moss, and percent lichen cover", which more accurately reflects the statistical analysis and results. We didn't originally include the p-values because we felt that in the mixed model analysis that was conducted, p values were not as meaningful to report as the best fitting model, which was selected through robust model selection. That said, we understand that p-values will clarify the results, and we have added them to the supplemental table (which was Table S3, but now is S2 because I adjusted to follow the order of the text).*

357: You described belowground C pools (to 1 m) as representing 92% of C pools; aboveground C pools should hence represent 8%, not 9%. I know it's a small (and statistically probably not significant) difference, but for consistency I think these figures should match.
*Corrected.*

Figure 3: Indicate in the figure caption that thaw depth was measured in July/August.
*Done.*

Table 4: Should the unit for C in permafrost cores not be kg C m-3?
*Yes, thank you for catching that mistake.*

Supplement tables are not well formatted. Please remove horizontal and vertical lines as appropriate. Some lines to separate e.g. sites (or possible cores within sites) are ok.
*These have been reformatted. I also added another Supplemental table (S5) with the data from the 15-m-deep cores; we previously had only included data from the 1-m cores.*